# Sodium–calcium exchanger 1 is the key molecule for urinary potassium excretion against acute hyperkalemia

Wakana Shoda[1], Naohiro Nomura [1]*, Fumiaki Ando[1], Hideaki Tagashira[2], Takahiro Iwamoto[2], Akihito Ohta[1]¤, Kiyoshi Isobe[1], Takayasu Mori[1], Koichiro Susa[1], Eisei Sohara[1], Tatemitsu Rai[1], Shinichi Uchida[1]

1 Department of Nephrology, Graduate School of Medical and Dental Sciences, Tokyo Medical and Dental University, Bunkyo, Tokyo, Japan, 2 Department of pharmacology, Faculty of Medicine, Fukuoka University, Fukuoka, Japan

¤ Current address: Department of Nephrology, Tokyo Metropolitan Komagome Hospital, Bunkyo, Tokyo, Japan

* nnomura.kid@tmd.ac.jp

**Data Availability Statement:** All relevant data are within the paper and its Supporting Information files.

## Abstract

The sodium ($Na^+$)-chloride cotransporter (NCC) expressed in the distal convoluted tubule (DCT) is a key molecule regulating urinary $Na^+$ and potassium ($K^+$) excretion. We previously reported that high-$K^+$ load rapidly dephosphorylated NCC and promoted urinary $K^+$ excretion in mouse kidneys. This effect was inhibited by calcineurin (CaN) and calmodulin inhibitors. However, the detailed mechanism through which high-$K^+$ signal results in CaN activation remains unknown. We used Flp-In NCC HEK293 cells and mice to evaluate NCC phosphorylation. We analyzed intracellular $Ca^{2+}$ concentration ($[Ca^{2+}]_{in}$) using live cell $Ca^{2+}$ imaging in HEK293 cells. We confirmed that high-$K^+$-induced NCC dephosphorylation was not observed without CaN using Flp-In NCC HEK29 cells. Extracellular $Ca^{2+}$ reduction with a $Ca^{2+}$ chelator inhibited high-$K^+$-induced increase in $[Ca^{2+}]_{in}$ and NCC dephosphorylation. We focused on $Na^+/Ca^{2+}$ exchanger (NCX) 1, a bidirectional regulator of cytosolic $Ca^{2+}$ expressed in DCT. We identified that NCX1 suppression with a specific inhibitor (SEA0400) or siRNA knockdown inhibited $K^+$-induced increase in $[Ca^{2+}]_{in}$ and NCC dephosphorylation. In a mouse study, SEA0400 treatment inhibited $K^+$-induced NCC dephosphorylation. SEA0400 reduced urinary $K^+$ excretion and induced hyperkalemia. Here, we identified NCX1 as a key molecule in urinary $K^+$ excretion promoted by CaN activation and NCC dephosphorylation in response to $K^+$ load.

## Introduction

Several epidemiological studies have reported that potassium ($K^+$) intake is inversely related to blood pressure [1, 2], the risk of cardiovascular disease, and mortality [3–5]. Evidence accumulated over the recent years has been drawing considerable attention to the importance of $K^+$

**Funding:** This work was supported by Grants-in-Aid for Scientific Research (KAKENHI) from Japan Society of the Promotion of Science (JSPS: https://www.jsps.go.jp/j-grantsinaid/index.html) and AMED (https://www.amed.go.jp/). The Grant Numbers by KAKENHI: JP18K19534(S.U), JP16K09642(T.R), JP16H05314 (E.S), JP18K15995 (N.N), JP18K15970 (K.I), JP17H06657(T.M), JP17H06656 and 18H08248 (F.A). The Grant Number by AMED: JPA17-108 and JP18058919 (S.U). This study was also supported in part by fund from the Central Research Institute of Fukuoka University (https://www.fukuoka-u.ac.jp/english/research/central_research/) under grant number No.171045 (T.I). The funders had no role in study design, data collection and analysis, decision to publish, or preparation of the manuscript.

**Competing interests:** The authors have declared that no competing interests exist.

intake. However, patients with kidney failure are particularly at risk of hyperkalemia. Therefore, the mechanism of $K^+$ excretion by the kidney needs to be elucidated.

Sodium ($Na^+$)-chloride ($Cl^-$) cotransporter (NCC), expressed along the apical membrane of the distal convoluted tubule (DCT) in the kidney, is essential for regulating urinary $K^+$ excretion and blood pressure. NCC does not directly transport $K^+$ but regulates $Na^+$ reabsorption in the DCT, controlling $Na^+$ delivery into the downstream nephron segments, where the epithelial sodium channel mediates electrogenic $Na^+$ reabsorption and $K^+$ is secreted through the renal outer medullary $K^+$ channel. Further, the study of two genetic diseases (Gitelman syndrome and pseudohypoaldosteronism type II) revealed the involvement of NCC in $K^+$ regulation. These diseases demonstrate that NCC activation and inactivation cause hyperkalemia [6] and hypokalemia [7], respectively.

NCC is phosphorylated and activated by Ste20-related proline/alanine-rich kinase (SPAK) regulated by Kelch-like protein 3 (KLHL3)-with-no-lysine kinase 4 (WNK4) cascade [8, 9]. The mechanism of NCC phosphorylation in low-$K^+$ conditions is known; low-$K^+$ stimulates WNK4–SPAK cascade, activating NCC [10, 11]. Terker et al. reported that low-$K^+$ hyperpolarized the basolateral membrane of DCT, causing the efflux of $Cl^-$ through the basolateral $Cl^-$ channels. The intracellular $Cl^-$ concentration ($[Cl^-]_{in}$) reduction activates the WNK signal, subsequently increasing NCC phosphorylation and activity. Experiments using genetically modified animal models confirmed that the Kir4.1/Kir5.1 $K^+$ [12, 13] and ClC-K/barttin $Cl^-$ channels [14] are involved in the low-$K^+$-induced reduction in $[Cl^-]_{in}$.

The mechanism of NCC inactivation following a high-$K^+$ load remains unknown. Rapid regulation of NCC following high-$K^+$ load differs from the response observed in the chronic phase. Castañeda-Bueno et al. showed an increase in NCC phosphorylation following the administration of a $K^+$-citrate diet for 2 days [15], whereas other reports have stated that KCl decreases NCC phosphorylation [10, 16]. Conversely, we previously reported that NCC was rapidly dephosphorylated by acute high-$K^+$ administration and this process was independent of the anion accompanying $K^+$ [17]. Thus, considering the acute and chronic phases separately is necessary to understand the effect of high-$K^+$ intake on NCC regulation. Aldosterone is another $K^+$ control system that responds to high $K^+$. However, a previous report showed that $K^+$-induced dephosphorylation of NCC is independent of aldosterone because NCC dephosphorylation occurred before there was an increase in aldosterone. In addition, NCC dephosphorylation was observed in aldosynthase-deficient mice [16]. In *ex vivo* mouse kidney slice experiments, Penton et al. showed that the inhibition of the plasma membrane $Cl^-$ flux using a $Cl^-$ channel blocker did not prevent NCC dephosphorylation in response to high-$K^+$ stimulation [18]. They concluded that a $Cl^-$-independent mechanism controls NCC dephosphorylation in response to high-$K^+$ intake and speculated about the involvement of protein phosphatase (PP) in this mechanism.

Several PPs, e.g. PP1 [19], Calcineurin (CaN, called PP2B [20]), PP4 [21], reportedly modulate NCC phosphorylation. In our previous study, we observed that the CaN inhibitor, tacrolimus, inhibited rapid $K^+$-induced NCC dephosphorylation and reduced urinary $K^+$ excretion in the acute phase [17]. Other studies have reported an increased abundance of NCC in mouse kidneys after treatment with CaN inhibitors [22, 23]. One study suggested that depolarization induced by $BaCl_2$ dephosphorylates NCC despite the presence of constitutively active SPAK in cultured cells and that tacrolimus inhibits NCC dephosphorylation [20]. These results suggest that CaN is a potent phosphatase dephosphorylating NCC under high-$K^+$ conditions. CaN is a $Ca^{2+}$- and calmodulin (CaM)-dependent serine/threonine PP comprising a catalytic CaN-A subunit that contains CaM-binding and autoinhibitory domains. CaN-A is constitutively bound to a regulatory CaN-B subunit possessing four EF-hand $Ca^{2+}$-binding domains [24]. CaN activation requires an increase in $[Ca^{2+}]_{in}$. Therefore, we hypothesized that an elevated

extracellular $K^+$ concentration ($[K^+]_{ex}$) increases $[Ca^{2+}]_{in}$ to activate CaN for rapid $K^+$ excretion in the kidney.

Herein, we used *in vitro*, *ex vivo* and *in vivo* models to identify the mechanism of $K^+$-induced rapid NCC dephosphorylation and urinary $K^+$ excretion.

## Materials and methods

### Plasmids

Human CaN-A, CaN-B, NCX1, and constitutively active CaN-A (CA-CaN-A) cDNAs were isolated using reverse transcription–polymerase chain reaction (RT–PCR) using human brain mRNA (Human Total RNA Master Panel II, BD Bioscience, Franklin Lakes, NJ, USA) and C57BL/6J mouse kidney mRNA as templates, respectively (primers shown in S1 Table in S1 File). CA-CaN-A was designed to be a truncated form of the catalytic Aα subunit, which lacks the autoinhibitory domain, and a portion of the calmodulin-binding domain yet retains the CaN-B-binding domain [25]. Subsequently, cDNAs were inserted into a T7-tagged pRK5 vector by Gibson assembly (New England Biolabs Inc, Ipswich, MA, USA). Site-directed mutagenesis was performed using PrimeSTAR MAX DNA polymerase (Takara Bio Inc., Shiga, Japan) to generate mutant CaN-B and NCX1. Each of the four EF-hand $Ca^{2+}$ binding sites (EF1–4) in CaN-B contains a single conserved glutamic acid (Glu/E) in the 12$^{th}$ position [26]. Because EF1 and EF2 are more important for CaN activity than sites EF3 and EF4 [26], we replaced the Glu with lysine (Lys/K) in both EF1 and EF2 (S1 Fig). SEA0400-insensitive NCX1 mutant (F213L NCX1) was constructed by replacing 213 phenylalanine (Phe/F) to leucine (Leu/L), as previously described [27].

### Cell culture and transfections

As previously described [28], HEK293 T-Rex cells, stably expressing NCC (Flp-In NCC HEK293), were cultured/selected in Dulbecco's modified Eagle's medium (Nacalai tesque, Kyoto, Japan), following which they were supplemented with 10% (v/v) fetal bovine serum, 100 units/ml penicillin, 15 μg/ml blasticidin, and 0.1 mg/ml hygromycin at 37˚C in a humidified 5% $CO_2$ incubator. Protein expression was induced using 10 μg/ml doxycycline for 24 h. To evaluate the NCC dephosphorylation, Flp-in NCC HEK293 cells were incubated in control ($K^+$ 3 mM) or high $K^+$ solution ($K^+$ 10 mM). The extracellular $K^+$ concentrations were determined based on a previous study [18]. Flp-In NCC HEK293 cells were transfected by the indicated amount of plasmid DNA with Lipofectamine 2000 reagent (Invitrogen, Carlsbad, CA, USA). For each transfection, the total amount of plasmid DNA was adjusted by adding empty vectors.

For NCX1 and SPAK small interfering RNA (siRNA) knockdown experiments, we used 30 pmol of SLC8A1 Human siRNA Oligo Duplex (CAT#: SR304429, OriGene, Rockville, MD, USA) and STK39 mouse siRNA TRIO (CAT#: SMF27A-2154; Cosmo Bio USA Co., Carlsbad, CA, USA). Flp-In NCC HEK293 cells were transfected with the siRNA using Lipofectamine RNAiMAX (Invitrogen, Carlsbad, CA, USA). Cells were incubated 48 h before use.

One micromolar 2-[4-[(2,5-difluorophenyl) methoxy]phenoxy]-5-ehoxyaniline (SEA0400; Cat # A15236, AdooQ Bioscience, Irvine, CA, USA) was used as an NCX1-specific inhibitor. One micromolar nifedipine (Cat # 21829, Sigma-Aldrich, St. Louis, MO, USA), 1 μM mibefradil (Cat # 15037, Cayman Chemical, MI, USA) and 100 μM nickel (II) chloride ($NiCl_2$, Cat #149–01041, Fuji Film Wako, Osaka, JAPAN), were used as a specific L-type $Ca^{2+}$ channel blocker, a specific T-type $Ca^{2+}$ blocker, and a T-type $Ca^{2+}$ channel and NCX1 inhibitor, respectively.

## Western blotting and immunofluorescence

Western blotting and immunofluorescence were performed as previously described [14, 29, 30]. The detail of the method is described in S1 Materials and Methods in S1 File. For Western blotting, the Flp-In NCC HEK293 cells were washed twice with phosphate-buffered saline and solubilized in lysis buffer. Kidney samples were homogenized and then the homogenates were centrifuged to separate entire kidney samples without the nuclear fraction, as either whole kidney lysates (600 g, supernatant) and crude membrane fraction (17000 g, pellet). The values of phosphorylated NCC bands were normalized to the total bands, whereas the values of other bands were normalized to actin. For immunofluorescence, kidneys were fixed by perfusion through the left ventricle with periodate lysine (0.2 M) and paraformaldehyde (2%) in PBS. Images were acquired using TCS SP8 confocal microscope (Leica Microsystems). Primary and secondary antibodies used in the present study are listed in Table 1. Specific bands obtained with phosphorylated SPAK antibody in HEK293T cells were confirmed by using siRNA knockdown (S2 Fig).

## Intracellular Cl⁻ measurement

Intracellular $Cl^-$ was measured as previously described [11]. Cells were plated in collagen-coated 96-well plates. Cells were loaded for 1 h with the $Cl^-$sensitive fluorescent dye, N-(ethoxycarbonylmethyl)-6-methoxyquinolinium bromide (MQAE) (Dojindo, Kumamoto, Japan) in Hank's Balanced Salt Solution (HBBS) (110 mM NaCl, 3 mM KCl, 1.2 mM $MgSO_4$, 1.8 mM $CaCl_2$, 4 mM Na acetate, 1 mM Na citrate, 6 mM D-glucose, 6 mM L-alanine, 1 mM $NaH_2PO_4$, 3 mM $Na_2HPO_4$, 25 mM $NaHCO_3$). After loading, cells were washed thrice, and fluorescence was measured on a FLUOstar OPTIMA-6 (BMG Labtech) using 350 nm excitation and 460 nm emission.

**Table 1. List of antibodies.**

| Protein | Host | Source [Reference] (Cat #/Lot#)) | Dilution | Dilution medium |
|---|---|---|---|---|
| **Primary antibodies** | | | | |
| pNCC (Ser 71) | Rabbit | [31] | 1:500 (WB) | TBST (WB) |
| pNCC (Thr 53) | Rabbit | [32] | 1:500 (WB, IF) | TBST (WB), 0.1% BSA in PBS (IF) |
| tNCC | Guinea pig | [29] | 1:500 (IF) | 0.1% BSA in PBS (IF) |
| tNCC | Rabbit | [29] | 1:500 (WB) | TBST (WB) |
| pSPAK (Ser 383) | Rabbit | [14],.S2 Fig | 1:500 | Can get signal *1 |
| T7-Tag monoclonal | Mouse | Novagen (69522–4) | 1:7500 | TBST |
| Pan calcineurin A | Rabbit | Cell Signaling (2614) | 1:500 | TBST |
| Calcineurin B | Rabbit | Abcam (ab154650) | 1:500 | TBST |
| Actin | Rabbit | Cytoskeleton (AAN01, Lot 121) | 1:1000 | TBST |
| Calbindin D28k | Mouse | Swant (#300, monoclonal) | 1:1500 (IF) | 0.1% BSA in PBS |
| **Secondary antibodies** | | | | |
| Rabbit IgG AP conjugate | N/A | Promega (S3738)/246053 | 1:7500 | 5% skim milk in TBST / Can get signal*1 (for SPAK antibodies) |
| Guinea pig IgG AP conjugate | N/A | Sigma (A2293)/10K4845 | 1:500 | 5% skim milk in TBST |
| Alexa-Rabbit IgG 488 | Goat | Molecular Probes (A11008/57099A) | 1:200 | 0.1% BSA in PBS |
| Alexa-Guinea pig IgG 546 | Goat | Molecular Probes (A11007/1073002) | 1:200 | 0.1% BSA in PBS |
| Alexa-Rabbit IgG 546 | Goat | Molecular Probes (A11010/1733163) | 1:200 | 0.1% BSA in PBS |
| Alexa-Mouse IgG 488 | Goat | Molecular Probes (A11001/50126A) | 1:200 | 0.1% BSA in PBS |
| Alexa-Guinea pig IgG 647 | Goat | Molecular Probes (A21450/2026140) | 1:200 | 0.1% BSA in PBS |

*1. Can Get Signal Immunoreaction Enhancer Solution (TOYOBO, Tokyo, Japan).

BSA: bovine serum albumin; IF: immunofluorescence; WB: Western blotting.

## Intracellular calcium assay

Intracellular calcium assay was performed as previously described [30]. The culture medium of the Flp-In NCC HEK293 cells was replaced by loading buffer containing 5 μg/ml Fluo 4-AM (Dojindo, Kumamoto, Japan), 1.25 mmol/l probenecid (Dojindo, Kumamoto, Japan), and 0.02% Pluronic F-127 (Dojindo, Kumamoto, Japan). Following incubation with 5 mM EGTA or 1 μM SEA0400 in the loading buffer for 1h at 37°C, the loading buffer was replaced by recording medium containing 1.25 mmol/l probenecid. NCX1 siRNA silencing was applied 48 h before loading buffer replacement. Following the administration of KCl (final concentration: 10 mM), the fluorescence intensities of Fluo 4 were quantified from five regions of interest using LSM 510 Meta confocal microscopy and the Zen 2009 software (Carl Zeiss, Oberkochen, Germany).

http://dx.doi.org/10.17504/protocols.io.baihicb6

## Animal experiments

All experiments were performed in accordance with the guidelines for animal research of the Tokyo Medical and Dental University, Tokyo, Japan, and The Animal Care and Use Committee of the Tokyo Medical and Dental University and Fukuoka University, Fukuoka, Japan, approved the protocol (A2018-183C3). Experiments were performed on male adult C57BL/6 mice (23–25 g of body weight), purchased from Japan SLC, Inc. All mice were housed under diurnal lighting conditions (light period: 8:00 a.m. to 8:00 p.m.). A high-$K^+$ solution ($K^+$ 1.7%, K-gluconate as dissolved in 2% sucrose) or a control solution (2% sucrose) was administered to mice via oral gavage (15 μl/g of body weight) as previously described [17].

SEA0400 was diluted in a mixture of 5% DMSO (Sigma-Aldrich, St. Louis, MO, USA) and 5% gum Arabic (Nacalai tesque, Kyoto, Japan). SEA0400 (10 mg/kg) or the vehicle solution was intraperitoneally injected into each mouse, 1 h before $K^+$ oral gavage. Kidneys were collected 15 min after oral gavage and protein samples were prepared for Western blotting and immunofluorescence.

Mice were housed in metabolic cages, and the urinary excretion study was performed as described previously [16]. Following treatment with SEA0400 and $K^+$ oral gavage, urine samples were collected every 30 min through either spontaneous voiding or bladder massage. Saline (1.5 ml) was loaded intraperitoneally 1 h before $K^+$ oral gavage, to prevent dehydration during the experiment. Urine data were analyzed using the DRI–CHEM analyzer (Fujifilm, Tokyo, Japan). At 120 min after $K^+$ oral gavage, blood was collected from the orbital venous plexus with the mice under anesthesia. Blood data were analyzed using iSTAT EC8t (Abbott, Inc., Abbott Park, IL).

## Ex vivo kidney slice experiment

Kidney slices were prepared as previously described [14, 33]. Before the harvest of both kidneys, mice were perfused at 17 ml/min using HBBS (110 mM NaCl, 3 mM KCl, 1.2 mM $MgSO_4$, 1.8 mM $CaCl_2$, 4 mM Na acetate, 1 mM Na citrate, 6 mM D-glucose, 6 mM L-alanine, 1 mM $NaH_2PO_4$, 3 mM $Na_2HPO_4$, 25 mM $NaHCO_3$) or $Ca^{2+}$-free-HBBS, under deep anesthesia. The kidneys were sliced (< 0.5 mm) using a microslicer (Natume Seisakusho Co., Ltd, Tokyo, Japan) and ice-cold HBBS. Following recovery in HBBS at room temperature for 20 min, the slices were transferred to the incubation chambers containing 3 mM $K^+$ or 10 mM $K^+$ with/without 5 mM EGTA. For thapsigargin and SEA0400 analysis, slices were pre-incubated in 100-μM thapsigargin (Sigma-Aldrich, St. Louis, MO, USA) and 50-μM SEA0400 for 30 min at room temperature before transferred to the incubation chambers. After incubation at 28°C for 30 min, slices were snap frozen in liquid nitrogen and processed for immunoblotting.

During the experiments, all solutions were continuously bubbled with 95% $O_2$ and 5% $CO_2$. (S3 Fig)

http://dx.doi.org/10.17504/protocols.io.baf9ibr6

### Electrophysiological calculation of the driving force of NCX1 in DCT

The magnitude of driving force of NCX ($E_{NCX}$) was calculated by using the following formula [34].

$$\mathbf{E_{NCX} = r \cdot E_{Na} - 2 \cdot E_{Ca} - (r - 2) \cdot E_m (mV)}$$

$E_{Na}$ and $E_{Ca}$ are the respective Nernst potential for $Na^+$ and $Ca^{2+}$, Em: the membrane potential calculated by Goldman–Hodgikin–Katz, r: the stoichiometric ratio, r = 3. Nernst equation:

$$\mathbf{V_{Eq}(mV) = (RT/zF) \times ln([X]_{out}/[X]_{in})}$$

Goldman–Hodgkin–Katz equation:

$$\mathbf{V_m = \frac{RT}{F} In\left(\frac{P_K[K^+]_o + P_{Na}[Na^+]_o + P_{Cl}[Cl^-]_i}{P_K[K^+]_i + P_{Na}[Na^+]_i + P_{Cl}[Cl^-]_o}\right)}$$

Ion concentrations and permeabilities are identical to those published [35–37].

$[K^+]_i$: 151.4 mM, $[Na^+]_o$: 144 mM, $[Na^+]_i$: 15.9 mM, $[Cl^-]_o$: 120.5 mM, $[Cl^-]_i$: 18.6 mM, $[Ca^{2+}]_o$: 1.2 mM, $[Ca^{2+}]_{in}$: 100 nM, $P_k$: 0.56, $P_{Na}$: 0.00, $P_{Cl}$: 0.19 (0.339 $P_K$).

### Statistical analysis

Data are presented as the means ± SEM. For multiple-group comparison, the mean value of "control, NK" group was set to "1", and all other ratios were compared to this. Two-way ANOVA using Tukey's test was performed to compare multiple groups, whereas the *t*-test was used to compare two groups. For all analyses, a *p*-value < 0.05 was considered statistically significant.

## Results

### Calcineurin is essential for K$^+$-induced NCC dephosphorylation

Hoorn et al. reported that tacrolimus treatment significantly increased phosphorylated NCC [22]. We have previously shown that a CaN inhibitor blocked the rapid NCC dephosphorylation in mouse kidneys [17]. In the present study, we generated constitutively active CaN-A (CA-CaN-A) by removing the autoinhibitory domain to investigate the direct contribution of CaN to NCC dephosphorylation *in vitro*. Following CA-CaN-A overexpression in Flp-In NCC HEK293, NCC was evidently dephosphorylated (Fig 1A).

It has been reported that HEK293T cells express low levels of endogenous CaN-B [38]. Unlike in the *in vivo* setting [17], this renders the evaluation of NCC dephosphorylation in HEK293 cells difficult. Consistent with previous reports, we observed higher endogenous CaN-A protein levels and extremely low endogenous CaN-B levels in Flp-In NCC HEK 293 cells (Fig 1B). Following transient CaN-B overexpression in Flp-In NCC HEK293 cells, we observed K$^+$-induced NCC dephosphorylation, which did not occur in CaN-B untransfected cells (Fig 1B). Therefore, Flp-In NCC overexpressing CaN-B was used in subsequent experiments unless otherwise noted. We observed that tacrolimus inhibited K$^+$-induced NCC dephosphorylation in Flp-In NCC HEK293 cells overexpressing CaNs (Fig 1C). These results confirmed that CaN is essential for K$^+$-induced NCC dephosphorylation.

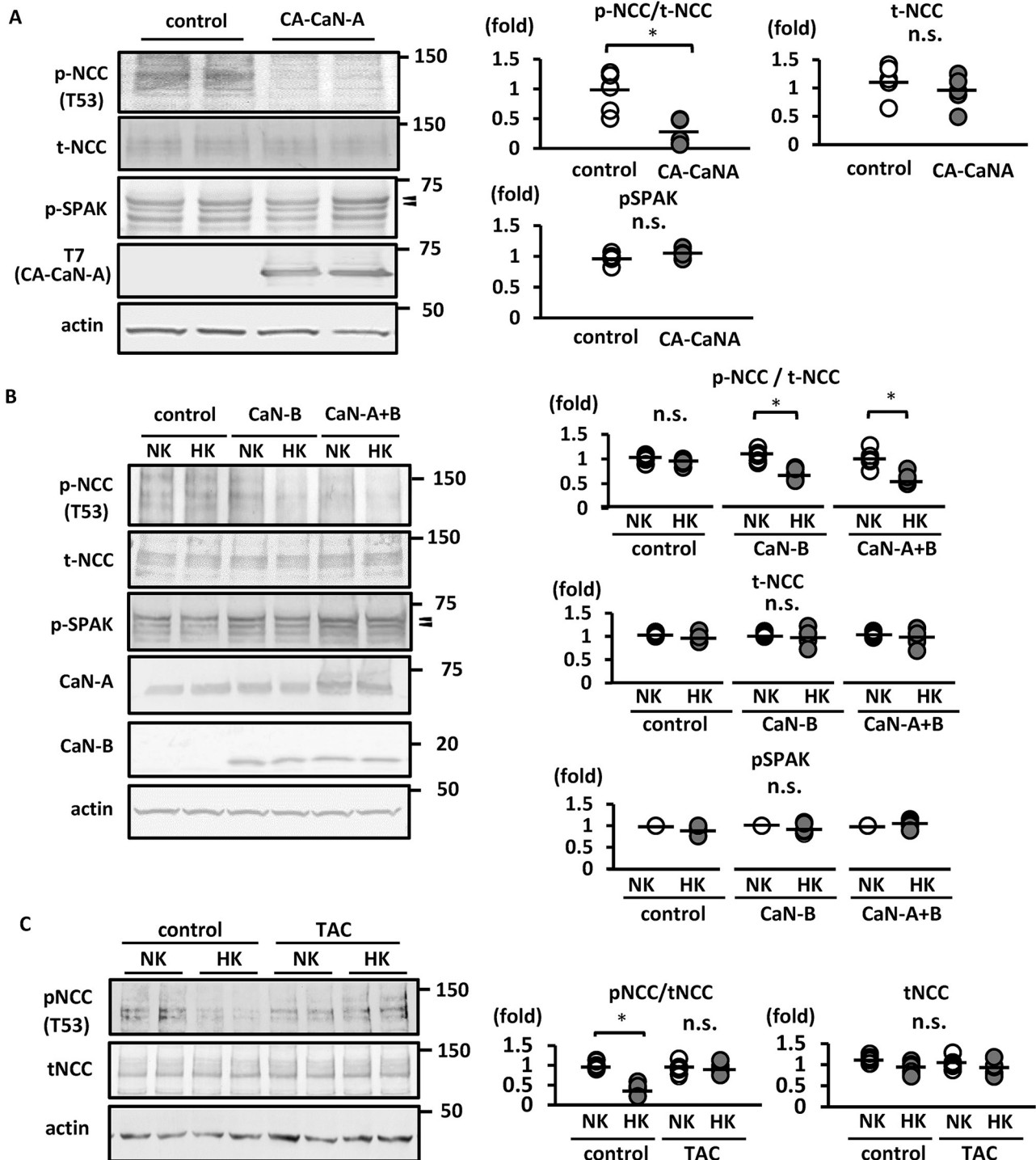

**Fig 1. Calcineurin activity is essential for K⁺-induced rapid NCC dephosphorylation in Flp-In NCC HEK293 cells.** A. (left) Representative immunoblots of CA-CaN-A overexpressed in Flp-In NCC HEK293 cells. CA-CaN-A significantly decreased the abundance of phosphorylated NCC. The abundance of phosphorylated SPAK shows no significant change in CA-CaN-A overexpression. (right) Quantitative analysis of the total and phosphorylated NCC ratio and phosphorylated SPAK in dot plots (n = 6). *p < 0.05 by unpaired t-test. B. (left) Representative immunoblots of Flp-In NCC HEK293 cells wherein CaN-B or CaN-A+CaN-B were overexpressed. Endogenous CaN-B was not expressed in Flp-In NCC HEK293 cells. Phosphorylated NCC level was decreased following high-K⁺ stimulation for 15 min in Flp-In NCC HEK293 cells, in which CaN-B or CaN-A+CaN-B were overexpressed. (right) Quantitative analysis of the total and phosphorylated NCC ratio and phosphorylated SPAK in dot plots (n = 6). *p < 0.05 by Tukey's test after multiple-way ANOVA. C. (left) Representative immunoblots of Flp-In NCC HEK293 cell treated with 1 μM tacrolimus. Treatment with tacrolimus for 1 h inhibited K⁺-induced NCC dephosphorylation in Flp-In NCC HEK293 cells. (right) Quantitative analysis of the total and

phosphorylated NCC ratio in dot plots (n = 6). *p < 0.05 by Tukey's test after two-way ANOVA. NCC, sodium–chloride cotransporter; pNCC, phosphorylated sodium–chloride cotransporter; tNCC, total sodium–chloride cotransporter; pSPAK, phosphorylated Ste20-related proline/alanine-rich kinase; NK, normal potassium (K+ 3 mM); HK, high potassium (K+ 10 mM); CaN-A, calcineurin A; CaN-B, calcineurin B; CA-CaN-A, constitutively active CaN-A; TAC, tacrolimus; n.s., not significant. Arrow heads indicate p-SPAK bands.

To investigate the involvement of the Cl–WNK–SPAK cascade in K+-induced NCC dephosphorylation, we evaluated SPAK phosphorylation and intracellular Cl− concentration ($[Cl^-]_{in}$) and observed no significant changes in pSPAK levels in HEK cells upon CA-CaN-A overexpression (Fig 1A). SPAK was not dephosphorylated under the high-K+ condition in Flp-In NCC HEK293 cells, even upon CaN overexpression (Fig 1B). $[Cl^-]_{in}$ was slightly higher under the high-K+ condition; however, the difference was not significant (S4 Fig). Together, these results conclude that K+-induced NCC dephosphorylation via CaN is independent of the Cl−-dependent WNK–SPAK–NCC signaling cascade, at least in the acute phase.

## Extracellular $Ca^{2+}$ is essential for K+-induced NCC dephosphorylation

Because CaN is a $Ca^{2+}$/CaM-dependent PP [39, 40], the increase in $[Ca^{2+}]_{in}$ is mandatory for CaN activation. To confirm the contribution of $Ca^{2+}$ to CaN activation and NCC dephosphorylation, mutant CaN-B, containing defective $Ca^{2+}$-binding sites, was transfected into Flp-In NCC HEK293 cells. CaN-B has four EF-hand $Ca^{2+}$ binding sites (EF1–4), which are important for CaN activation [41, 42]. The CaN-B mutant was constructed wherein EF1 and EF2 were selectively disrupted (S1 Fig). As shown in Fig 2A, K+-induced NCC dephosphorylation was not observed in Flp-In HEK293 cells overexpressing the mutant CaN-B protein (Fig 2A). We analyzed $[Ca^{2+}]_{in}$ following K+ administration using live cell $Ca^{2+}$ imaging with Fluo 4-AM in the HEK293 cells; subsequent results revealed an evident increase in $[Ca^{2+}]_{in}$ (Fig 2B). The K+-induced increase in $[Ca^{2+}]_{in}$ was completely inhibited by EGTA, an extracellular $Ca^{2+}$ chelator; the ensuing removal of $[Ca^{2+}]_{ex}$ using EGTA inhibited K+-induced NCC dephosphorylation (Fig 2C). We conducted *ex vivo* kidney slice experiments as previously described [14, 33], to confirm the *in vitro* findings. We observed that NCC was dephosphorylated in high-K+ medium in wild-type mouse kidney slices. In kidney slices pre-incubated in $[Ca^{2+}]_{ex}$-free medium with EGTA, K+-induced NCC dephosphorylation was significantly suppressed (Fig 2D). We evaluated $Ca^{2+}$ release from the endoplasmic reticulum (ER) using thapsigargin in *ex vivo* kidney slices. Kidney slices were pre-incubated in 100-µM thapsigargin solution; this depleted $Ca^{2+}$ stored in the ER. Thapsigargin did not inhibit K+-induced NCC dephosphorylation (S5 Fig). These results suggested that a K+ load promoted $Ca^{2+}$ influx from the extracellular space, leading to the increase in $[Ca^{2+}]_{in}$ and NCC dephosphorylation.

## Reverse-mode NCX plays a role in the $Ca^{2+}$ influx pathway

We hypothesized that a $Ca^{2+}$ transporter in DCT cells are involved in the increase of $Ca^{2+}$ influx following a high-K+ load. The $Na^+$/$Ca^{2+}$ exchanger (NCX) 1 is a major $Ca^{2+}$ transporter in DCT cells, mediating $Ca^{2+}$ entry following cell depolarization. Therefore, we focused on NCX1 as a potential candidate in the $Ca^{2+}$-influx pathway. NCX is a bidirectional transporter regulated by the membrane potential and transmembrane gradients of $Na^+$ and $Ca^{2+}$ [43, 44] and expressed on the basolateral membrane of DCT cells [45]. Under cell depolarization, NCX may operate in the "reverse-mode" to mediate $Ca^{2+}$ entry [46, 47].

Because NCX1 is primarily expressed in the late DCT and connecting tubule (CNT) [48] and NCC is expressed primarily in the early DCT, double immunofluorescence was initially performed to confirm NCX1 and NCC co-expression in mouse kidneys. We observed that NCX1 and NCC were evidently co-expressed (S6 Fig). Subsequently, we demonstrated NCX1

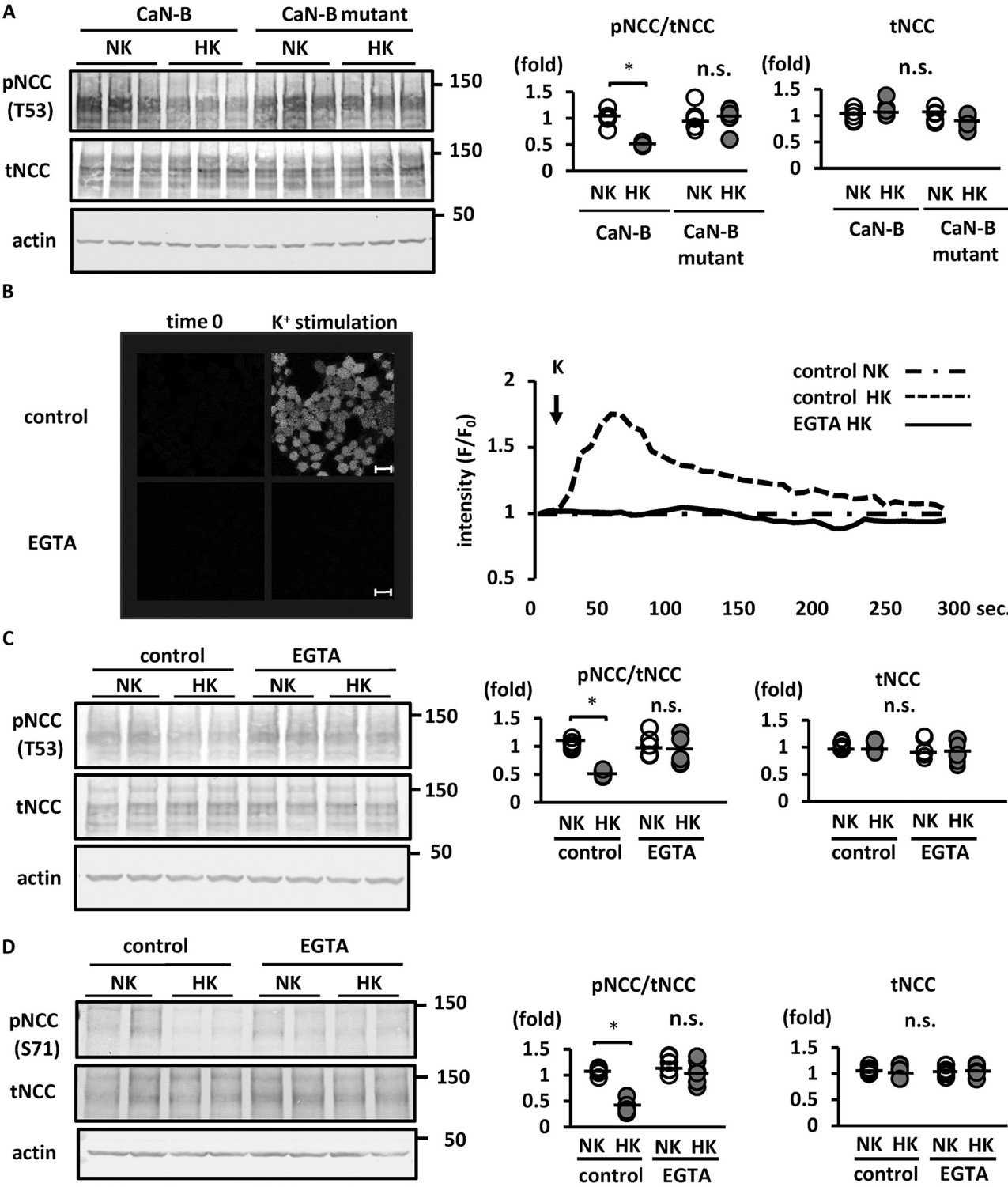

**Fig 2. Ca²⁺ influx after high-K⁺ stimulation is essential for NCC dephosphorylation in Flp-In NCC HEK293 cells.** A. (left) Representative immunoblots of Ca²⁺-binding-deficient CaN-B (CaN-B mutant) overexpressed in Flp-In NCC HEK293 cells. Overexpression of Ca²⁺-binding-deficient CaN-B suppressed K⁺-induced NCC dephosphorylation. (right) Quantitative analysis of the total and phosphorylated NCC ratio in dot plots (n = 6). *p < 0.05 by Tukey's test after two-way ANOVA. B. (left) Representative images of Fluo 4 intensity 30 s after K⁺ administration. K⁺ (10 mM final concentration) was added to Flp-In NCC HEK293 cells in an EGTA-containing medium or control medium. The increased Fluo 4 fluorescence intensity following addition of K⁺ was inhibited in the EGTA-containing medium. Scale bars, 20 μm. (right) Representative time-course of Fluo 4 fluorescence intensity. The x and y axes indicate time and Fluo 4 fluorescence intensity, respectively. C. (left) Representative immunoblots of Flp-In NCC HEK293 cells in 5 mM EGTA-containing medium. The high-K⁺-induced reduction in phosphorylated NCC level was inhibited in the EGTA-

containing medium. (right) Quantitative analysis of the total and phosphorylated NCC ratio in dot plots (n = 6). *p < 0.05 by Tukey's test after two-way ANOVA. D. (left) Representative immunoblots of mouse kidney slices *ex vivo*. The high-K⁺-induced reduction in phosphorylated NCC level was inhibited in the EGTA-containing medium. (right) Quantitative analysis of the total and phosphorylated NCC ratio in dot plots (n = 6). *p < 0.05 by Tukey's test after two-way ANOVA. NCC, sodium–chloride cotransporter; pNCC, phosphorylated sodium–chloride cotransporter; tNCC, total sodium–chloride cotransporter; NK, normal potassium (K⁺ 3 mM); HK, high potassium (K⁺ 10 mM); CaN-B, calcineurin B; n.s., not significant.

expression in Flp-In HEK 293 cells, using RT–PCR and Western blotting (Fig 3A). We treated the cells with 1 μM SEA0400 (a selective inhibitor of reverse-mode NCX1 [49]) for 1 h before stimulating with 10 mM K⁺ and observed that SEA0400 evidently inhibited the increase in $[Ca^{2+}]_{in}$ following a K⁺ load, as observed in live cell $Ca^{2+}$ imaging (Fig 3B). We confirmed that $NiCl_2$ (an inhibitor of T-type $Ca^{2+}$ channel and NCX) inhibited K⁺-induced NCC dephosphorylation (S7 Fig). Nifedipine (specific L-type $Ca^{2+}$ channel blocker), mibefradil (specific T-type $Ca^{2+}$ channel blocker) were used to inhibit other $Ca^{2+}$ channels. Treatments with nifedipine and mibefradil did not inhibit neither the increase in $[Ca^{2+}]_{in}$ nor NCC dephosphorylation following a K⁺ load (Fig 3B, S7 Fig). Then, we clarified that SEA0400 effectively inhibited K⁺-induced NCC dephosphorylation (Fig 3C). To confirm these results we showed in cultured cells, we performed *ex vivo* experiments as well which reflects *in vivo* function. we observed K⁺-induced NCC dephosphorylation was suppressed in kidney slices incubated in SEA0400 (Fig 3D). To verify that this effect of SEA0400 is specific for NCX1, we constructed F213L NCX1, insensitive to SEA0400. We previously reported that the SEA0400-insensitve mutant is useful to assess the pharmacological significance of NCX1 inhibition by performing a hypoxia/reoxygenation-induced cell damage experiment [27]. We overexpressed either wild-type NCX1 or F213L mutant NCX1 in Flp-In HEK293 cells and treated the cells with SEA0400 1 h before K⁺ administration. SEA0400, which presumably could inhibit both endogenous NCX1 and transfected wild-type NCX1, suppressed K⁺-induced NCC dephosphorylation in cells overexpressing wild-type NCX1 (S8 Fig). Conversely, in the cells overexpressing F213L NCX1, K⁺-induced NCC dephosphorylation was not inhibited by SEA0400 (S8 Fig). This suggests that NCX1 inhibition by SEA0400 was responsible for the suppression of K⁺-induced NCC dephosphorylation.

We also used siRNA to suppress NCX1 expression in order to confirm the role of NCX1 in high-K⁺-induced NCC dephosphorylation. The suppression of NCX1 expression using siRNA was confirmed both by Western blotting and quantitative RT–PCR (Fig 3A and S9 Fig). K⁺-induced NCC dephosphorylation was not observed following the siRNA silencing of NCX1 expression in Flp-In HEK293 cells (Fig 4A). In addition, NCX1 silencing suppressed the increase in $[Ca^{2+}]_{in}$ following a K⁺ load in Flp-In HEK293 cells, as observed by live cell $Ca^{2+}$ imaging (Fig 4B).

## Blockade of NCX1 inhibited rapid NCC dephosphorylation after a K⁺ load in mouse kidneys

To investigate the role of NCX1 in K⁺-induced NCC dephosphorylation and urinary K⁺ excretion *in vivo*, we administered 10 mg/kg SEA0400 intraperitoneally to adult C57BL/6 mice 1 h before 1.7% K⁺ oral gavage. The SEA0400 dosage was determined according to a previous study [49]. SEA0400-treated mice appeared normal, acting in a manner similar to vehicle-treated mice. We observed that SEA0400 evidently inhibited rapid NCC dephosphorylation following a K⁺ load by Western blotting and immunofluorescence (Figs 5 and 6A). To investigate the part of nephron segments wherein NCC was specifically dephosphorylated after K⁺ load, phosphorylated NCC were co-stained with calbindin, a marker of late DCT. We confirmed that calbindin overlapped with NCX1 as shown in a previous study [48] (S10 Fig). We observed that the phosphorylated NCC was slightly retained in the kidneys with K⁺ load; this

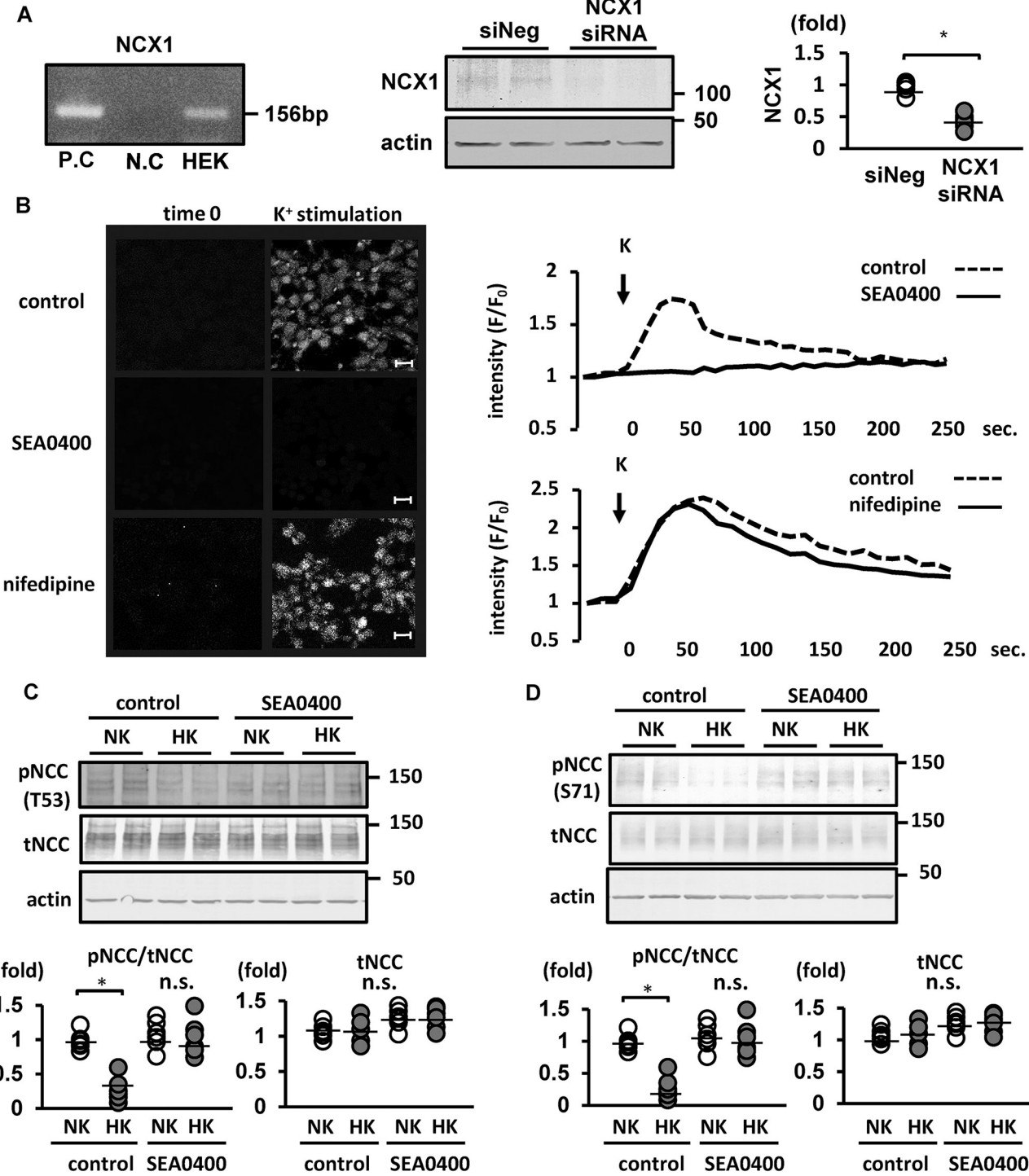

**Fig 3. Pharmacological suppression of sodium–calcium exchanger (NCX) 1 inhibited the increase in $[Ca^{2+}]_{in}$ and sodium–chloride cotransporter (NCC) dephosphorylation following high-$K^+$ stimulation in Flp-In NCC HEK293 cells.** A. (left) mRNA expression of NCX1 was observed in Flp-In NCC HEK293 cells. Samples containing distilled water and human brain cDNA were used as negative and positive controls, respectively. P.C., positive control; N.C., negative control. (middle) Representative immunoblots of NCX1 in Flp-In NCC HEK293 cells with NCX1 siRNA. (right) Quantitative analysis of protein levels of NCX1 in Flp-In NCC HEK293 cells with NCX1 siRNA (n = 6). NCX1 protein expression was significantly reduced after NCX1 silencing with siRNA (n = 6). siNeg, negative control with siRNA, * represents significant differences at p <0.05 using an unpaired t-test. B. (left) Representative images of Fluo-4 fluorescence intensity 30 s after $K^+$ administration. A 10 mM final concentration of $K^+$ was added to Flp-In NCC HEK293 cells with/without 1 μM SEA0400. Increased Fluo-4 fluorescence intensity following $K^+$ administration was inhibited through treatment with SEA0400. Scale bars, 20 μm. (right) Representative time-course of Fluo-4 fluorescence intensity. The x and y axes indicate time and Fluo-4 fluorescence intensity, respectively. Nifedipine (1 μM) was used as the negative control. C. (upper) Representative immunoblots of a Flp-In NCC HEK293 cell

treated with 1 μM SEA0400. Treatment with SEA0400 inhibited K$^+$-induced NCC dephosphorylation. (lower) Quantitative analysis of the total and phosphorylated NCC in dot plots (n = 6). *p < 0.05 by Tukey's test after two-way ANOVA. n.s., not significant. NK, normal potassium (K$^+$ 3 mM); HK, high potassium (K$^+$ 10 mM). D. (upper) Representative immunoblots of mouse kidney slices *ex vivo*. Treatment with 50 μM SEA0400 inhibited K$^+$-induced NCC dephosphorylation. (lower) Quantitative analysis of the total and phosphorylated NCC in dot plots (n = 6). *p < 0.05 by Tukey's test after two-way ANOVA. n.s., not significant. NK, normal potassium (K$^+$ 3 mM); HK, high potassium (K$^+$ 10 mM).

retained NCC was mainly observed in the early DCT wherein calbindin was not stained (Fig 5). In addition, we used NCX1$^{+/-}$ KO mice, in which NCX1 expression is approximately 50% of that reported in wild-type mice [50]. In NCX1$^{+/-}$ KO mice, K$^+$-induced NCC dephosphorylation was not evident (S11 Fig).

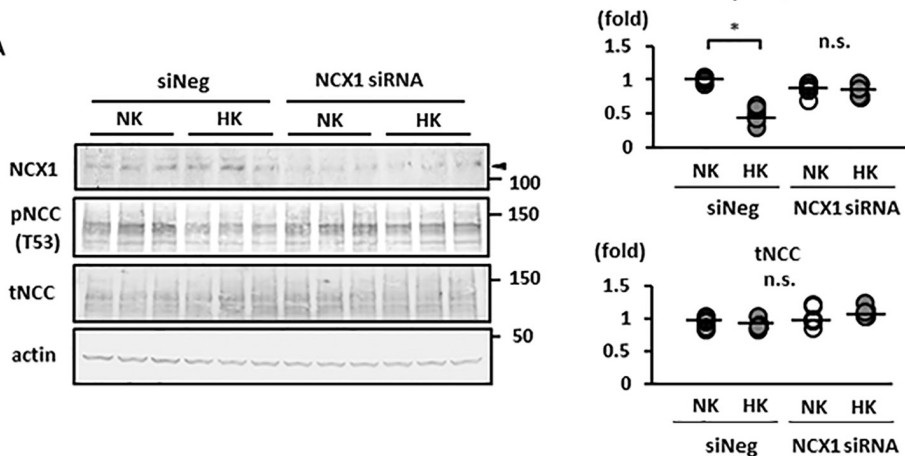

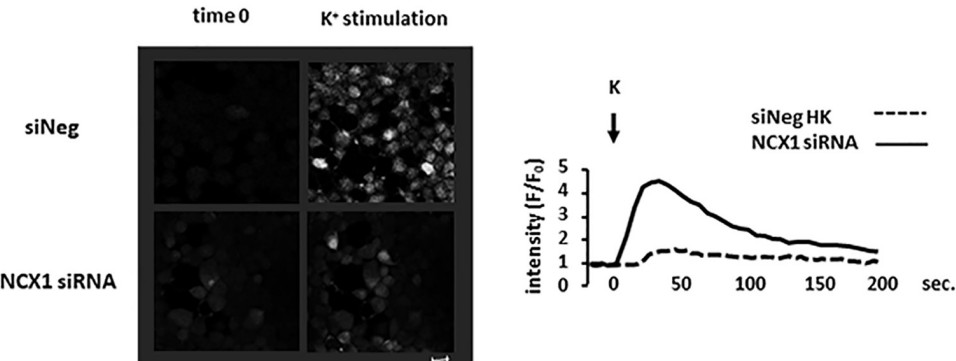

**Fig 4. Sodium–calcium exchanger (NCX) 1 silencing with small interfering RNA (siRNA) suppressed K$^+$-induced sodium–chloride cotransporter (NCC) dephosphorylation.** A. (left) Representative immunoblots of Flp-In NCC HEK293 cells with NCX1 siRNA, which inhibited K$^+$-induced NCC dephosphorylation. (right) Quantitative analysis of the total and phosphorylated NCC in dot plots (n = 6). * represents significant differences at p <0.05 using Tukey's test after a two-way ANOVA. n.s., not significant. B. (left) Representative images of Fluo-4 fluorescence intensity 30 s after K$^+$ administration. K$^+$ was added to Flp-In NCC HEK293 cells with NCX1 siRNA and the negative control siRNA (siNeg). The increase in Fluo-4 fluorescence intensity following K$^+$ administration was attenuated in NCX1-silenced cells. Scale bars, 20 μm. (right) Representative time-course of Fluo-4 fluorescence intensity. The x and y axes indicate time and Fluo 4 fluorescence intensity, respectively. NK, normal potassium (K$^+$ 3 mM); HK, high potassium (K$^+$ 10 mM).

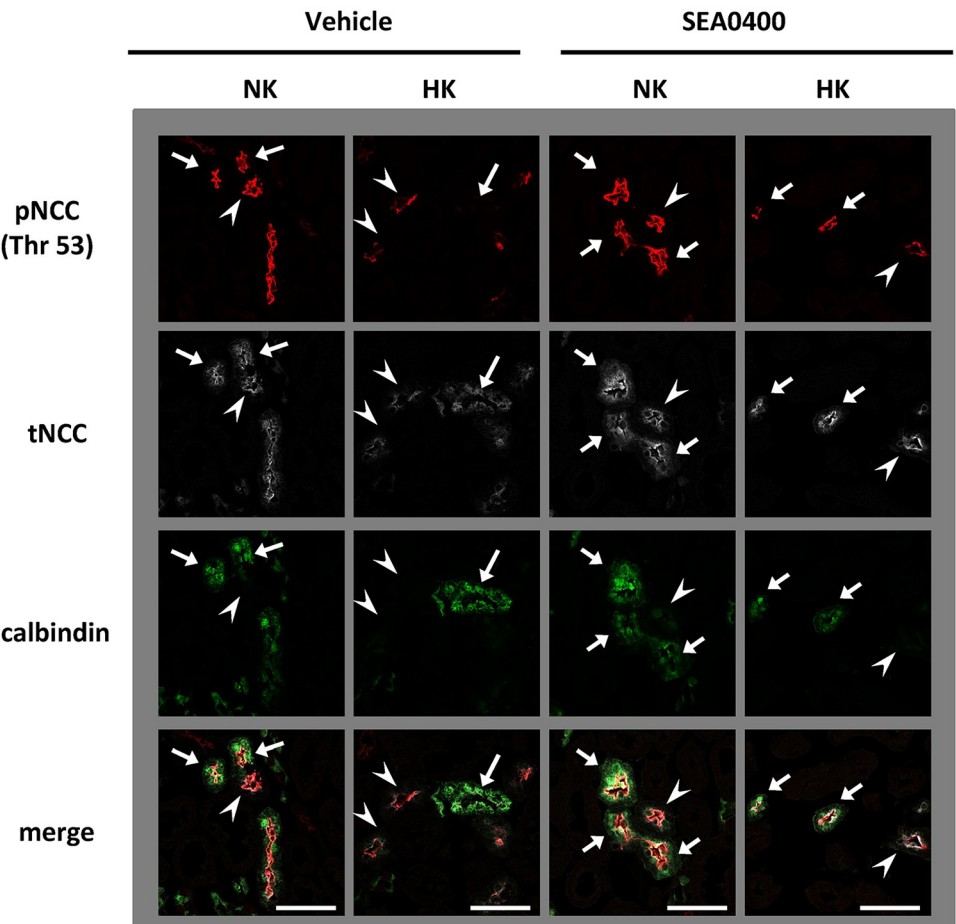

**Fig 5. K⁺-induced NCC dephosphorylation is observed mainly in late DCT.** Representative immunofluorescences. Calbindin was used as a marker of late DCT. NCC expressed early DCT (arrow heads) and late DCT (arrows). K⁺-induced NCC dephosphorylation was observed mainly in late DCT. The K⁺-induced dephosphorylation of NCC in late DCT was inhibited by SEA0400 treatment. Red: pNCC (Thr 53), white: tNCC, green: Calbindin. Scale bars indicate 50 μm.

To investigate the physiological contribution of NCX1 to NCC-related kaliuresis, we analyzed urinary $K^+$ excretion and blood $K^+$ level under acute $K^+$ load. SEA0400-treated mice demonstrated significantly lower urinary $K^+$ excretion than vehicle-treated mice at 30 and 90 min following a high-$K^+$ load (Fig 6B). In SEA0400-treated mice, $Na^+$ and $Cl^-$ excretion were significantly low and exhibited a low tendency, respectively (Fig 6B). Blood $K^+$ level at 120 min following a high-$K^+$ load was significantly higher in SEA0400-treated mice than in vehicle-treated counterparts (Table 2). This indicates that NCX1 plays a role in $K^+$-induced NCC dephosphorylation and urinary $K^+$ excretion, similar to the $[Ca^{2+}]_{ex}$ influx pathway in the acute phase of $K^+$ loading. Because the primary role of NCX1 is $Ca^{2+}$ reabsorption, we investigated urinary $Ca^{2+}$ excretion after oral $K^+$ loading. Urine $Ca^{2+}$ excretion was similar in $K^+$-administered and control mice (S12 Fig). We speculated the involvement of a compensational system for $Ca^{2+}$ reabsorption.

## Discussion

Herein, we identified the mechanism of $Ca^{2+}$ signaling in DCT cells and verified the mechanism of high-$K^+$-induced rapid NCC dephosphorylation. We observed that $Ca^{2+}$ influx

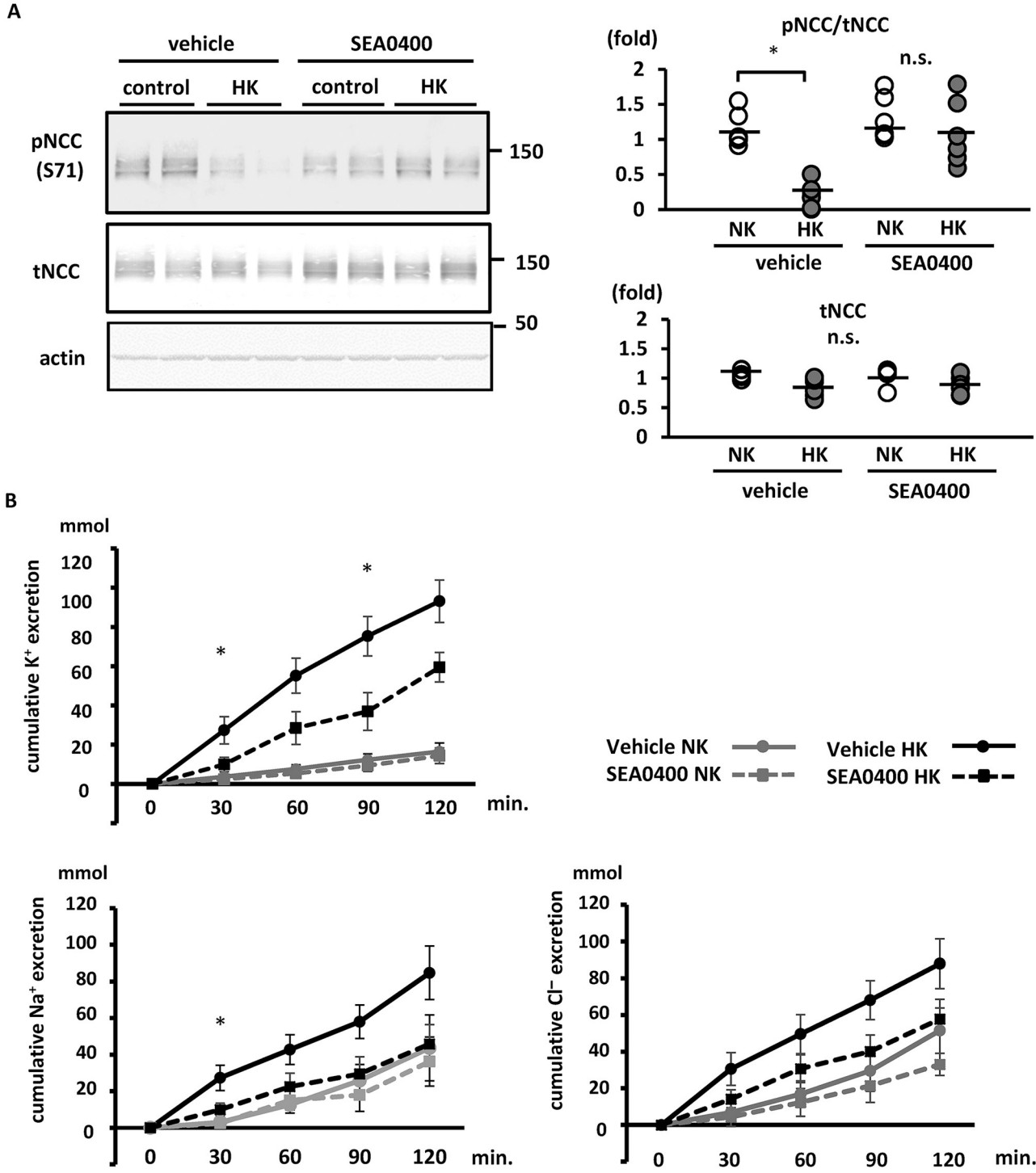

**Fig 6. Sodium–calcium exchanger (NCX) 1 contributes to the rapid NCC dephosphorylation and kaliuresis following K⁺ oral administration in mice.** A. (left) Representative immunoblots of total and phosphorylated NCC in mouse kidneys treated with SEA0400. SEA0400 or vehicle was intraperitoneally injected, 1 h before K⁺ oral gavage. Kidneys were collected 15 min after oral gavage. The rapid NCC dephosphorylation following the K⁺ load was significantly inhibited by treatment with SEA0400. (right) Quantitative analysis of blots from both the total and phosphorylated NCC in mouse kidneys shown in dot plots (n = 6). $^*p < 0.05$ by Tukey's test after two-way ANOVA. B. Cumulative analysis of urinary K⁺, Na⁺ and Cl⁻ excretion following K⁺ oral gavage (n = 6). Mice were pre-treated with SEA0400 1 h before K⁺ oral gavage. Following K⁺ administration, urine was collected every 30 min. In the SEA0400-treated mice, urinary K⁺ excretion was significantly suppressed at 30 and 90 min after K⁺ administration. The urinary Na⁺ excretion was significantly suppressed at 30 min after K⁺ administration. Urinary Cl⁻ excretion was lower at all-time points in SEA0400-treated mice; however, the difference was not statistically significant. Mean ± standard error of the mean. $^*$represents significant differences at $p < 0.05$ using Tukey's test after a two-way ANOVA at each time point.

**Table 2. Blood data for the mice 120 min after K⁺ oral gavage.**

| | | control (N = 6) | High-K⁺ (N = 6) |
|---|---|---|---|
| **Na (mmol/l)** | | | |
| | vehicle | 148 ± 1 | 149 ± 1 |
| | SEA0400 | 148 ± 1 | 149 ± 1 |
| **K (mmol/l)** | | | |
| | vehicle | 4.8 ± 0.2 | 5.5 ± 0.2* |
| | SEA0400 | 4.7 ± 0.1 | 6.5 ± 0.4* |
| **Cl (mmol/l)** | | | |
| | vehicle | 112 ± 1 | 109 ± 2 |
| | SEA0400 | 110 ± 2 | 113 ± 1 |
| **pH** | | | |
| | vehicle | 7.27 ± 0.02 | 7.27 ± 0.01 |
| | SEA0400 | 7.27 ± 0.03 | 7.28 ± 0.03 |

Data shown are mean ± SEM. Statistical significance between each groups were assessed by two-way analysis of variance using Tukey's test.

* $p < 0.05$. Na, sodium; K, potassium; Cl, chloride.

occurred following K⁺ stimulation and was inhibited by EGTA treatment and NCX1 suppression. Previously, we showed that CaN rapidly dephosphorylated NCC in response to high-K⁺ intake [17]. It is hypothesized that a high-K⁺ condition reverses the action of NCX, leading to $Ca^{2+}$ influx; this increases $[Ca^{2+}]_{in}$, activates CaN, and eventually leads to NCC dephosphorylation. The proposed signaling pathway is summarized in Fig 7. Our *in vivo* experimental results

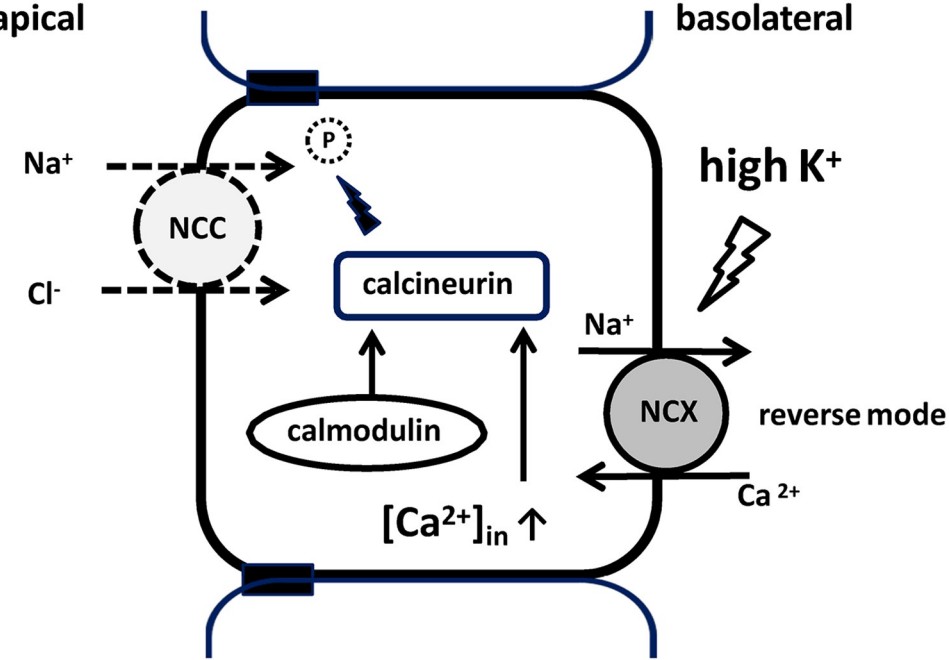

**Fig 7. K⁺-induced NCC dephosphorylation via CaN: The proposed mechanism.** High plasma K⁺ level depolarizes the cell membrane potential, leading to $Ca^{2+}$ influx through reverse-mode NCX and consequently an increase in $[Ca^{2+}]_{in}$. The increase in $[Ca^{2+}]_{in}$ activates CaN, which dephosphorylates NCC. Na⁺, sodium; $Ca^{2+}$, calcium; K⁺, potassium; P, phosphorylation; $[Ca^{2+}]_{in}$, intracellular calcium concentration; NCC, sodium–chloride cotransporter; NCX, sodium–calcium exchanger.

support the proposed mechanism, showing that SEA0400 treatment reduces urinary $K^+$ excretion following high-$K^+$ load.

The adrenal gland also controls $K^+$ balance. In adrenal zona glomerulosa cells, increase in plasma $[K^+]$ leads to depolarization, opening voltage-dependent $Ca^{2+}$ channels (VDCCs) and stimulating the production and release of aldosterone [51]. To our knowledge, there is no report conclusively demonstrating physiological role of VDCCs in DCT cells. In DCT, $Ca^{2+}$ reabsorption occurs via transcellular pathway. The transient receptor potential cation channel subfamily V member 5 (TRPV5) and TRPV6 on the apical membrane reabsorb $Ca^{2+}$ into DCT cells. Basolateral $Ca^{2+}$ extrusion was mediated by NCX and plasma membrane $Ca^{2+}$ ATPase (PMCA) in DCT cell [52]. In DCT, TRPV5 is a major $Ca^{2+}$ channel involved in $Ca^{2+}$ influx. Membrane depolarization has not been reported to affect the TRPV5 activity. Conversely, hyperpolarization increased TRPV5 activity, promoting $Ca^{2+}$ influx into the cells [53]. Therefore, TRPV5 is unlikely to be involved in mediating high-$K^+$-induced $Ca^{2+}$ influx as a $Ca^{2+}$ transporter. NCX may increase $Ca^{2+}$ influx through the reverse mode in response to increased $[K^+]_{ex}$. NCX1 has been reported to mediate the electrogenic stoichiometry of ion-exchange ($3Na^+$:$Ca^{2+}$), operating in forward ($Ca^{2+}$-efflux) or reverse ($Ca^{2+}$-influx) mode. The mode depends on the intracellular or extracellular $[Na^+]$ and $[Ca^{2+}]$ as well as on the membrane potential [43, 54]. Reverse-mode NCX has been implicated in the $Ca^{2+}$ influx pathway in other cell types [55, 56]. Drumm et al. reported that the depolarization of interstitial cells of Cajal at a high-$K^+$ condition increased intracellular $Ca^{2+}$ waves by $Ca^{2+}$ influx via reverse-mode NCX [57]. Therefore, following a high-$K^+$ load, the reverse-mode NCX1 is a potential $Ca^{2+}$ influx pathway in DCT cells. To confirm whether membrane depolarization in response to a change in $[K^+]_{ex}$ is sufficient to drive $Ca^{2+}$ entry, we calculated the magnitude of NCX driving force ($E_{NCX}$) using a formula assigned with ion concentrations and solute permeability in the DCT, as described previously [34, 36, 52, 58]. We observed that high-$K^+$ condition reversed the $E_{NCX}$ (S13 Fig). Although computationally estimated, these results support the hypothesis and suggest that the high-$K^+$ condition leads NCX1 as a reverse mode at least in standard conditions in the DCT.

Mammalian NCX proteins have three isoforms—NCX1 [59], NCX2 [60], and NCX3 [61]. NCX1 is the major isoform and is widely expressed in the heart, kidney, brain, arteries, and other organs. Conversely, NCX2 and NCX3 expressions are the highest in the brain and skeletal muscle [59]. In the kidney, NCX1 is predominantly localized in the basolateral membranes of late DCTs and CNT [48, 52, 62]. Gotoh et al. suggested that NCX2 is expressed in the DCT [45]. However, an RNA-deep-sequencing study of microdissected renal tubules did not corroborate this finding [63], indicating that NCX2 expression in the DCT is much lower than NCX1 expression.

In a previous study [45], NCX1 heterozygous KO mice and SEA0400-treated mice did not show significant changes in urinary $K^+$ excretion, compared with wild-type and vehicle-treated mice, respectively. NCX2 inhibition promoted a significant increase in urinary $K^+$ excretion at 24 h. During our short observation period, NCX1 inhibition using SEA0400 did not suppress urinary $K^+$ excretion in control mice without a $K^+$ load, suggesting that NCX1 does not contribute to $K^+$ excretion in a steady-state condition. Following a rapid increase in $[K^+]_{ex}$, NCX1 functions in reverse mode, contributing to the increase in urinary $K^+$ excretion. NCC dephosphorylation is more rapid than plasma aldosterone elevation in response to $K^+$ intake [16]. Therefore, the mechanism of rapid NCC dephosphorylation plays a protective role against hyperkalemia in the acute phase of $K^+$ intake.

Melnikov et al. [23] showed that the administration of cyclosporine leads to Gordon syndrome-like symptoms such as pNCC elevation, hyperkalemia, and hypertension. Therefore, one can speculate that inhibition of NCX1 leading to inhibition of CaN could potentially cause

Gordon syndrome-like symptoms. However, to verify the role of CaN and NCX1 in the pathogenesis of Gordon syndrome, suppressing NCX1 for a relatively long period is necessary. Although mutations of Cul3, KLHL3, and WNKs are known as the cause of Gordon syndrome, the NCX–CaN–NCC regulation system is independent of the Cul3–KLHL3–WNK regulation cascade. Further investigation is required to evaluate whether the NCX1–CaN–NCC regulation system is related to Gordon syndrome. To the best of our knowledge, there are no reports of Gordon syndrome caused by mutations in NCX1 or CaN.

NCX1 is expressed in various tissues; therefore, future studies should determine whether our novel finding (i.e., a high- $K^+$-induced $Ca^{2+}$ influx via the NCX1 reverse-mode) is observable in other tissues, e.g., considering the high NCX1 expression in the heart [64], the involvement of NCX1 in high-$K^+$-induced arrhythmia should be investigated. An association between NCX1 mutations and certain diseases (including QT prolongation [65, 66] and Kawasaki disease [67]) has recently been suggested. Studies should investigate the presence of urinary $K^+$ excretion abnormalities in response to an acute $K^+$ load in such patients.

Herein, we discovered that NCX1, in addition to CaN, is important for rapid NCC dephosphorylation in response to $K^+$ load in the kidney. Depolarization of cells reverses the mode of NCX function, leading to $Ca^{2+}$ influx into the cells. The increased $[Ca^{2+}]_{in}$ activates CaN, leading to rapid NCC dephosphorylation. This mechanism is involved in urinary $K^+$ excretion in the acute phase of $K^+$ intake.

## Supporting information

**S1 Fig. Positions of $Ca^{2+}$-binding sites in calcineurin B.** A $Ca^{2+}$-binding-deficient CaN-B mutant was constructed using site-directed mutagenesis. Glutamic acid (E) in the 12th position of the 1st and 2nd EF-hand $Ca^{2+}$-binding sites (shown in red characters) was replaced with lysine (K).
(TIF)

**S2 Fig. Confirmation of phospho-specific-Ste20-related proline/alanine-rich kinase (SPAK) antibody in HEK293 cells.** Immunoblots of phosphorylated SPAK in HEK293 cells with SPAK siRNA silencing and the negative control (siNeg). The disappearance of bands from cell lysates with SPAK siRNA silencing confirms the specificity of the antibody (shown with arrows).
(TIF)

**S3 Fig. Image of the *ex vivo* kidney slice experiment system.** Kidneys were sliced into sections of <0.5 mm thickness. All solutions were continuously bubbled with 95% $O_2$ and 5% $CO_2$. Details are described in the Materials and Methods section.
(TIF)

**S4 Fig. Intracellular $[Cl^-]$ analysis after high-$K^+$ administration (n = 4).** Intracellular $[Cl^-]$ was analyzed using the $Cl^-$-sensing dye MQAE. High-$K^+$ administration of 10 mM $K^+$ did not show a significant change in intracellular $[Cl^-]$ compared with normal $K^+$ administration of 3 mM $K^+$. *represents significant differences at p <0.05 using an unpaired *t*-test. NK, normal potassium; HK, high potassium.
(TIF)

**S5 Fig. $K^+$-induced NCC dephosphorylation was not inhibited by treatment with thapsigargin in mouse kidney slices *ex vivo*.** (left) Representative immunoblots of mouse kidney slices with 100 μM thapsigargin. The high-$K^+$-induced reduction in the level of phosphorylated NCC was not inhibited in thapsigargin treatment. (right) Quantitative analysis of the total and

phosphorylated NCC ratio in dot plots (n = 6). *p <0.05 by Tukey's test after two-way ANOVA. NK, normal potassium; HK, high potassium. n.s., not significant.
(TIF)

**S6 Fig. Colocalization of NCX1 and NCC in a mouse kidney.** Double immunofluorescence of total NCC (green) and NCX1 (red) in mouse kidneys. NCC and NCX1 were co-localized in the distal convoluted tubule. Scale bars: 50 μm.
(TIF)

**S7 Fig. Evaluation of the effect of $Ca^{2+}$ channel blockers on $K^+$-induced NCC dephosphory-lation and $Ca^{2+}$ influx.** A. Evaluation of high-$K^+$-induced NCC dephosphorylation in Flp-In NCC HEK293 cells treated with 1 μM nifedipine, 1 μM mibefradil, and 100 μM $NiCl_2$. Normal $K^+$ and high $K^+$ are 3 mM and 10 mM, respectively. (right) Representative immunoblots. Although $NiCl_2$ inhibited $K^+$-induced NCC dephosphorylation, mibefradil and nifedipine did not. (right) Quantitative analysis of the total and phosphorylated NCC ratio in dot plots (n = 6). *represents significant differences at p <0.05 using Tukey's test after a multiple-way ANOVA. $NiCl_2$, nickel chloride; NK, normal potassium; HK, high potassium; n.s., not signifi-cant. B. The influx of $Ca^{2+}$ after high-$K^+$ stimulation in mibefradil treatment in Flp-In NCC HEK293 cells. (left) Representative images of Fluo 4 intensity 30 s after $K^+$ administration. $K^+$ (10 mM final concentration) was added to Flp-In NCC HEK293 cells 1 h after mibefradil treat-ment. The increased Fluo 4 fluorescence intensity following addition of $K^+$ was not inhibited by mibefradil treatment. Scale bars, 20 μm. (right) Representative time-course of Fluo 4 fluo-rescence intensity. The x and y axes indicate time and Fluo 4 fluorescence intensity, respec-tively.
(TIF)

**S8 Fig. The inhibitory effect of SEA0400 on $K^+$-induced NCC dephosphorylation was recovered by overexpression of SEA0400-insensitive mutant NCX1.** (left) Representative immunoblots of total and phosphorylated NCC in Flp-In NCC HEK293 cells overexpressing an NCX1 mutant (F213L) and wild-type NCX1. The inhibition of $K^+$-induced NCC dephos-phorylation with SEA0400 treatment was recovered in the cells overexpressing F213L NCX1. NK, normal potassium ($K^+$ 3 mM); HK, high potassium ($K^+$ 10 mM). (right) Quantitative analysis of the total and phosphorylated NCC ratio in column graphs (n = 6). *p < 0.05 by Tukey's test after two-way ANOVA.
(TIF)

**S9 Fig. mRNA and protein expression of sodium-calcium exchanger (NCX) 1 in Flp-In sodium–chloride cotransporter (NCC) HEK293 cells after siRNA silencing.** Relative mRNA expression of NCX1 in Flp-In NCC HEK293 cells following NCX1 siRNA silencing was significantly decreased compared with that in cells with negative control siRNA. n = 6. Means ± standard errors of the mean. *represents significant differences at p <0.05 using an unpaired t-test.
(TIF)

**S10 Fig. Representative immunofluorescence images of sodium–calcium exchanger (NCX) 1 and calbindin.** Confirmation of the localization of NCX1 and calbindin in wild-type mouse kidney. Red: NCX1, Green: calbindin. Scale bars indicate 50 μm and 200 μm in high and low magnification images, respectively.
(TIF)

**S11 Fig. Rapid NCC dephosphorylation following a high-$K^+$ load was not significant in sodium–calcium exchanger (NCX) 1$^{+/-}$ knockout (KO) mice.** (left) Immunoblots of total

NCC and phosphorylated NCC in NCX$^{+/-}$ KO mice and wild-type mice. A rapid decrease in the level of phosphorylated NCC after K$^+$ administration was not evident in NCX$^{+/-}$ KO mice. (right) Quantitative analysis of total and phosphorylated NCC in NCX$^{+/-}$ KO mice shown in column graphs (n = 3). $^*$ represents significant differences at p $<$0.05 using an unpaired t-test. (TIF)

**S12 Fig. Cumulative Ca$^{2+}$ excretion in wild-type mice following K$^+$ administration.** Urinary Ca$^{2+}$ excretion was not different between mice administered high K$^+$ and those administered normal K$^+$ (n = 6). Means ± SEM. $^*$p $<$0.05 by unpaired t-test at each time point. (TIF)

**S13 Fig. Electrophysiological calculation analysis of sodium-calcium exchanger (NCX) 1 driving force in mouse distal convoluted tubule cells at a variety of K$^+$ concentrations.** NCX drives in forward-mode and reverse-mode under low- and high-K$^+$ conditions, respectively. E$_{NCx}$, driving force of NCX. (TIF)

**S1 Raw image.** (TIF)

**S2 Raw image.** (TIF)

**S3 Raw image.** (TIF)

**S1 File.** (DOCX)

## Acknowledgments

We thank Prof. Dario Alessi (University of Dundee, United Kingdom) for providing HEK-293 T-Rex cell lines stably expressing NCC. We also thank Shintaro Mandai and Yuri Takeda for help in the experiments.

## Author Contributions

**Conceptualization:** Wakana Shoda, Naohiro Nomura, Eisei Sohara, Tatemitsu Rai, Shinichi Uchida.

**Data curation:** Hideaki Tagashira.

**Formal analysis:** Wakana Shoda.

**Funding acquisition:** Naohiro Nomura.

**Investigation:** Wakana Shoda, Naohiro Nomura, Fumiaki Ando.

**Methodology:** Wakana Shoda, Naohiro Nomura.

**Project administration:** Wakana Shoda, Naohiro Nomura, Fumiaki Ando, Eisei Sohara, Tatemitsu Rai, Shinichi Uchida.

**Resources:** Hideaki Tagashira, Takahiro Iwamoto, Akihito Ohta.

**Supervision:** Fumiaki Ando, Takahiro Iwamoto, Akihito Ohta, Kiyoshi Isobe, Takayasu Mori, Koichiro Susa, Eisei Sohara, Tatemitsu Rai, Shinichi Uchida.

**Validation:** Wakana Shoda, Naohiro Nomura, Fumiaki Ando, Hideaki Tagashira, Takahiro Iwamoto.

**Visualization:** Wakana Shoda, Naohiro Nomura.

**Writing – original draft:** Wakana Shoda.

**Writing – review & editing:** Wakana Shoda, Naohiro Nomura, Fumiaki Ando, Kiyoshi Isobe, Takayasu Mori, Koichiro Susa, Eisei Sohara, Tatemitsu Rai, Shinichi Uchida.

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
