## [Decision Letter · Decision Letter 0]

14 Apr 2020

PONE-D-19-36071

Sodium–calcium exchanger 1 is the key molecule for urinary potassium excretion against acute hyperkalemia

PLOS ONE

Dear Mr. Nomura,

Thank you for submitting your manuscript to PLOS ONE. After careful consideration, we feel that it has merit but does not fully meet PLOS ONE’s publication criteria as it currently stands. Therefore, we invite you to submit a revised version of the manuscript that addresses the points raised during the review process. As you will see, Reviewer 1 would appreciate more physiological information in your in vivo studies, Reviewer 2 asks questions about the suitability of your in vivo model using a NCX-1 inhibitor. Reviewer 3, although generally supportive of your manuscript, has several concerns regarding the presentation of your results and the choice of some parameters.

We would appreciate receiving your revised manuscript by May 29 2020 11:59PM. To enhance the reproducibility of your results, we recommend that if applicable you deposit your laboratory protocols in protocols.io, where a protocol can be assigned its own identifier (DOI) such that it can be cited independently in the future. For instructions see: http://journals.plos.org/plosone/s/submission-guidelines#loc-laboratory-protocols

We look forward to receiving your revised manuscript.

Kind regards,

Jean-Claude Dussaule

Academic Editor

PLOS ONE

Journal Requirements:

1. PLOS ONE now requires that authors provide the original uncropped and unadjusted images underlying all blot or gel results reported in a submission’s figures or Supporting Information files. This policy and the journal’s other requirements for blot/gel reporting and figure preparation are described in detail at https://journals.plos.org/plosone/s/figures#loc-blot-and-gel-reporting-requirements and https://journals.plos.org/plosone/s/figures#loc-preparing-figures-from-image-files. When you submit your revised manuscript, please ensure that your figures adhere fully to these guidelines and provide the original underlying images for all blot or gel data reported in your submission. See the following link for instructions on providing the original image data: https://journals.plos.org/plosone/s/figures#loc-original-images-for-blots-and-gels.

Reviewers' comments:

Reviewer's Responses to Questions

**Comments to the Author**

1. Is the manuscript technically sound, and do the data support the conclusions?

Reviewer #1: Yes

Reviewer #2: Yes

Reviewer #3: Yes

2. Has the statistical analysis been performed appropriately and rigorously? 

Reviewer #1: Yes

Reviewer #2: I Don't Know

Reviewer #3: Yes

3. Have the authors made all data underlying the findings in their manuscript fully available?

Reviewer #1: Yes

Reviewer #2: Yes

Reviewer #3: Yes

4. Is the manuscript presented in an intelligible fashion and written in standard English?

Reviewer #1: Yes

Reviewer #2: Yes

Reviewer #3: Yes

5. Review Comments to the Author

Reviewer #1: The study by Wakana Shoda et al. describes the role of the Na/Ca-exchanger 1 and of calcineurin in the K+-dependent inhibition of NCC (through dephosphorylation). They first confirm previous results showing that calcineurin dephosphorylates NCC and then complete this observation by showing that increase of intracellular Ca is a requirement for this effect. Then they show, in vivo and ex vivo (on kidney slices) that NCX1 inhibition alters the adaptation to acute K load by maintaining NCC in its phosphorylated state. The data presented support the conclusion and few points coild be added to improve the manuscript.

1/ The authors should clearly mention in the figure legends the concentrations of K+ they used in the conditions Normal and High K. Could you also comment on why you choose 10 mM KCl for the study on cells ?

2/ Could you include the expression of NCX1 in HEK cells that are depicted in the Supplemental Information S6 into the Fig 3, do you have a western blot in addition to the PCR ?

3/ The modification of activity of NCC, in vivo, is only related to its phosphorylation state (Figure 6 and 7). A test of thiazide sensitivity after SEA0400 and K treatment could be helpful to strengthen the idea that SEA0400 interfere with NCC activity in vivo. The Supplemental information 11 and 12 are the same figures, please include the urine Na and Cl excretion in the figure 7 to outline that the treatment with NCX1 inhibitor leads to Na retention.

4/Could you provide plasma aldosterone concentration for the 4 groups of treated mice? If NCC is not inhibited as it should in the case of NCX1 inhibition + K loading, which induces hypekalemia, and decrease of K and Na excretion, the mice should be hypertensive. I invite the authors to measure the BP of their mice because they may have here a model of Gordon syndrome (fHHt). The authors should discuss the possibility of CaN and NCX1 to participate to this disease.

Reviewer #2: The manuscript entitled « sodium-calcium exchanger 1 is a key molecule for urinary potassium excretion against acute hyperkalemia » by Wakana Shoda et al. shows that NCX1 is a key regulator of urine potassium excretion in response to increased (serum/extracellular) potassium levels, promoting the activation of calcineurin and downstream NCC dephosphorylation. Interestingly, thapsigargin inhibiting calcium release from ER did not inhibit NCC dephophorylation, suggesting that only the calcium flux from extracellular space was involved in the response to potassium load. Of note, cells were exposed in vitro to very high potassium levels (10mM), this is understandable but makes in vivo models indispensable to confirm the in vitro experiments.

In vitro studies are well designed and the concept is very interesting.

Main concern: in vivo studies are based upon pharmacological inhibition of NCX1. A model of acute potassium load in NCX-1 knock-out (heterozygous) mice would be more convincing. These mice seem to be available in Japan.

Minor concerns

Page 6: “In our previous study, we observed that the CaN inhibitor, tacrolimus....”: reference 22 seems inappropriate. Did the authors mean to reference 17 ?

Page 23: reference 47 seems inappropriate too

Reviewer #3: The manuscript PONE-D-19-36071, entitled ‘Sodium–calcium exchanger 1 is the key molecule for urinary potassium excretion against acute hyperkalemia’, by Shoda and co-authors, describes the characterisation of the crucial role played by the Na-Ca exchanger NCX1 in the regulation of the activity of the Na-Cl cotransporter NCC in the distal nephron during a K-load.

Overall, the results described here support the hypotheses and the conclusions drawn. However, I have the following comments.

Major comments:

Material and Methods – Animal experiment : why were the kidneys collected 15 minutes after the oral K gavage for protein extraction? The metabolic data were collected every 30 minutes after the oral gavage. Why is it different?

Results :

(1) Figure 1 : Calcineurin is essential for K+-induced NCC dephosphorylation

- I don’t understand the result and relevance of the experiment describing the consequence of the overexpression of CaN-A on NCC phosphorylation. How could CaN-A work without CaN-B, which is barely expressed in HEK293 cells, as explained by the authors in the next paragraph?

Conversely, why did the authors overexpress CaN-A when transfecting CaN-B is sufficient to induce NCC dephosphorylation (Fig 1B)?

(2) Figure 2D: why did the authors quantify pNCC-S71 in kidney slices and mice rather than NCC-p53, used in vitro?

In addition, the figures would be easier to read if the system in which the experiments are conducted were indicated on the figure.

(3) Figure S11: why did the authors quantify the pNCC/tNCC ratio rather than pNCC and tNCC separately like in the other experiments?

(4) Table 2: ‘Blood K+ level at 120 min following a high-K+ load was significantly higher in SEA0400-treated mice than in vehicle-treated counterparts (Table 2)’.

In their previous study, using tacrolimus to inhibit calcineurin, the authors did not observe a change in plasma potassium in tacrolimus-treated animals even though NCC dephosphorylation was inhibited. Could the authors comment on the difference in plasma K between tacrolimus- and SEA0400-treated animals?

(5) Have the authors studied the role of NCX1 after a longer exposure to high K ?

Minor comments:

- Abstract : the authors should remove the following sentence, which is not at the correct place : ‘The mice were housed in metabolic cages for urine sample collection’.

- Introduction, page 6 : the reference’s number in the following sentence is incorrect: ‘In our previous study, we observed that the CaN inhibitor, tacrolimus, …urinary K+ excretion in the acute phase[22]’. The authors should check all the other references’ number.

- Results: Figures 3 and 5 should be merged.

6. PLOS authors have the option to publish the peer review history of their article (what does this mean?). If published, this will include your full peer review and any attached files.

Reviewer #1: No

Reviewer #2: No

Reviewer #3: Yes: Juliette Hadchouel

---

## [Author Response · Author response to Decision Letter 0]

23 May 2020

Reviewer #1: The study by Wakana Shoda et al. describes the role of the Na/Ca-exchanger 1 and of calcineurin in the K+-dependent inhibition of NCC (through dephosphorylation). They first confirm previous results showing that calcineurin dephosphorylates NCC and then complete this observation by showing that increase of intracellular Ca is a requirement for this effect. Then they show, in vivo and ex vivo (on kidney slices) that NCX1 inhibition alters the adaptation to acute K load by maintaining NCC in its phosphorylated state. The data presented support the conclusion and few points coild be added to improve the manuscript.

1. The authors should clearly mention in the figure legends the concentrations of K+ they used in the conditions Normal and High K. Could you also comment on why you choose 10 mM KCl for the study on cells?

Thank you for this important suggestion. We added information about the K+ concentrations to each figure legend. 

The K+ concentration for the high K+ treatment was chosen based on a previous study [1]. These authors demonstrated that 5 mmol/L of extracellular [K+] as a control was already high enough for suppression of NCC phosphorylation. We tested 5, 7, and 10 mM K+ solutions as options for high K+ for HEK cells in our preliminary experiments, and we found that a comparison between K+ 3 mM and K+ 10 mM was best to see differences in NCC dephosphorylation. Therefore, we chose 3 mM K+ as the control and 10 mM K+ as the high K solution. We have added this information to the Materials and Methods section (P 9, L 1).

2. Could you include the expression of NCX1 in HEK cells that are depicted in the Supplemental Information S6 into the Fig 3, do you have a western blot in addition to the PCR ?　

Thank you for this comment. In our previous manuscript, we showed a Western blot which demonstrated the expression of NCX1 in HEK cells in Previous Fig S9. NCX1 specific bands were confirmed using siRNA silencing. We have integrated the RT–PCR and Western blotting data for NCX1 into a revised Fig 3A.

3. The modification of activity of NCC, in vivo, is only related to its phosphorylation state (Figure 6 and 7). A test of thiazide sensitivity after SEA0400 and K treatment could be helpful to strengthen the idea that SEA0400 interfere with NCC activity in vivo. The Supplemental information 11 and 12 are the same figures, please include the urine Na and Cl excretion in the figure 7 to outline that the treatment with NCX1 inhibitor leads to Na retention.

We sincerely apologize for uploading S11 Fig. and S12 Fig. as duplicate information. We have corrected the mistake and included urine Na and Cl excretion in Figure 7 following the kind instruction of Reviewer #1.

As for the thiazide test, we hypothesized that the response to thiazide should be suppressed by high-K-induced NCC dephosphorylation, whereas the response to thiazide should be maintained after SEA0400 treatment because SEA0400 inhibits K-induced NCC dephosphorylation. We have preliminary data to support the hypothesis. However, we have not had enough time to perform additional experiments during a situation where there are some restrictions on conducting experiments. We have shown the preliminary data in this response letter. The response to thiazide was quite blunted after acute K load (urinary K and Na excretion control vs. TZD in the vehicle group). Compared with this blunted response, the response to thiazide was larger in the SEA0400-treated group (R1 Fig.). This result indicates that the suppression of NCC activity after exposure to the K load was inhibited by the SEA0400 treatment. 

4. Could you provide plasma aldosterone concentration for the 4 groups of treated mice? If NCC is not inhibited as it should in the case of NCX1 inhibition + K loading, which induces hypekalemia, and decrease of K and Na excretion, the mice should be hypertensive. I invite the authors to measure the BP of their mice because they may have here a model of Gordon syndrome (fHHt). The authors should discuss the possibility of CaN and NCX1 to participate to this disease.

Thank you for the important comment. We agree that a discussion about aldosterone and blood pressure is important. However, our institution is now in emergency response to COVID-19 and is not permitting new animal experiments to be conducted. We are very sorry about this. Alternatively, we would like to briefly comment about aldosterone and blood pressure.

As for aldosterone, the acute NCC dephosphorylation after exposure to the K load is believed to be independent of aldosterone. Sorensen et al. [2] showed that plasma aldosterone levels were not significantly increased 15 min after oral KCl gavage. They also showed that K-induced NCC dephosphorylation was observed in aldosterone synthase knockout mice as well [2]. Therefore, we believe that the mechanism of K-induced acute NCC dephosphorylation, in which NCX1 and calcineurin are involved, is independent of the aldosterone system. Based on these observations, we have added some information to the Introduction (P6.L5). 

As for the evaluation of blood pressure, it is possible that NCX1 suppression led to the increase in blood pressure. Indeed, previous studies have shown that the administration of calcineurin inhibitors led to hypertension after several days [3,4]. According to these previous studies, to verify whether NCX1 suppression can increase blood pressure, it is necessary to suppress NCX1 for a relatively long period. However, in the present study, we are focusing on the acute phase after increased K load, because we know that a chronic K load induces a more complicated physiological response. Therefore, the evaluation of blood pressure may not be suitable in the present study.

As for the participation of NCX1 and CaN in the pathogenesis of Gordon syndrome, to the best of our knowledge, there are no reports to show that the mutation of NCX1 or CaN causes Gordon syndrome. Mutations of Cul3, KLHL3, or WNKs are known as the causes of Gordon syndrome. The NCX-CaN-NCC regulation system is independent of the Cul3-KLHL3-WNK regulation cascade, as we demonstrated. Further investigation is required to evaluate whether the NCX1-CaN-NCC regulation system is related to Gordon syndrome. We have accordingly added some information in the Discussion to highlight these points (P34, L9).

References:

1. Penton D, Czogalla J, Wengi A, Himmerkus N, Loffing-Cueni D, Carrel M, et al. Extracellular K(+) rapidly controls NCC phosphorylation in native DCT by Cl(-) -dependent and -independent mechanisms. J Physiol. 2016.

2. Sorensen MV, Grossmann S, Roesinger M, Gresko N, Todkar AP, Barmettler G, et al. Rapid dephosphorylation of the renal sodium chloride cotransporter in response to oral potassium intake in mice. Kidney Int. 2013;83(5):811-24. 

3. Hoorn EJ, Walsh SB, McCormick JA, Fürstenberg A, Yang CL, Roeschel T, et al. The calcineurin inhibitor tacrolimus activates the renal sodium chloride cotransporter to cause hypertension. Nat Med. 2011;17(10):1304-9.

4. Melnikov S, Mayan H, Uchida S, Holtzman EJ, Farfel Z. Cyclosporine metabolic side effects: association with the WNK4 system. Eur J Clin Invest. 2011;41(10):1113-20.

Reviewer #2: The manuscript entitled « sodium-calcium exchanger 1 is a key molecule for urinary potassium excretion against acute hyperkalemia » by Wakana Shoda et al. shows that NCX1 is a key regulator of urine potassium excretion in response to increased (serum/extracellular) potassium levels, promoting the activation of calcineurin and downstream NCC dephosphorylation. Interestingly, thapsigargin inhibiting calcium release from ER did not inhibit NCC dephophorylation, suggesting that only the calcium flux from extracellular space was involved in the response to potassium load. Of note, cells were exposed in vitro to very high potassium levels (10mM), this is understandable but makes in vivo models indispensable to confirm the in vitro experiments.

In vitro studies are well designed and the concept is very interesting.

Main concern: in vivo studies are based upon pharmacological inhibition of NCX1. A model of acute potassium load in NCX-1 knock-out (heterozygous) mice would be more convincing. These mice seem to be available in Japan.

Thank you for the insightful comment. We tested NCX1+/− KO mice (P28. L3) and found that NCX1 expression is approximately 50% of that reported in wild-type mice. In NCX1+/− KO mice, K-induced NCC dephosphorylation was significantly suppressed (revised S11 Fig.). We believe that this result supports our findings. However, verification is needed by conducting further experiments with kidney-specific NCX1 knockout mice.

Minor concerns

Page 6: “In our previous study, we observed that the CaN inhibitor, tacrolimus....”: reference 22 seems inappropriate. Did the authors mean to reference 17 ?　　

Page 23: reference 47 seems inappropriate too.　

We sincerely apologize for the incorrect citations. We have checked all the reference numbers and corrected them.

Reviewer #3: The manuscript PONE-D-19-36071, entitled ‘Sodium–calcium exchanger 1 is the key molecule for urinary potassium excretion against acute hyperkalemia’, by Shoda and co-authors, describes the characterisation of the crucial role played by the Na-Ca exchanger NCX1 in the regulation of the activity of the Na-Cl cotransporter NCC in the distal nephron during a K-load.

Overall, the results described here support the hypotheses and the conclusions drawn. However, I have the following comments.

Major comments:

Material and Methods – Animal experiment : why were the kidneys collected 15 minutes after the oral K gavage for protein extraction? The metabolic data were collected every 30 minutes after the oral gavage. Why is it different?

We collected kidneys 15 min after K administration to avoid the effects of aldosterone and other secondary effects. The regulation of NCC dephosphorylation could be different between acute and chronic K load, as described in the Introduction section (P6, L5). As for the metabolic data, previously, Sorensen et al. [1] showed that K-induced dephosphorylation of NCC continued until 2 h after K-load. Therefore, we followed urine data until 2 h after K administration. We collected urine every 30 min because 15 min is too short a period to collect urine for mice. 

Results :

(1) Figure 1 : Calcineurin is essential for K+-induced NCC dephosphorylation

- I don’t understand the result and relevance of the experiment describing the consequence of the overexpression of CaN-A on NCC phosphorylation. How could CaN-A work without CaN-B, which is barely expressed in HEK293 cells, as explained by the authors in the next paragraph?

Conversely, why did the authors overexpress CaN-A when transfecting CaN-B is sufficient to induce NCC dephosphorylation (Fig 1B)?

Thank you for the insightful comments. Fig. 1A was intended to show the importance of the activity of CaN to dephosphorylate NCC using “constitutively active CaN-A (CA-CaN-A).” CA-CaN-A is an active form of CaN, which was designed to be a truncated form of the catalytic Aα subunit and lacks the autoinhibitory domain, meaning that CA-CaN-A is always active even without CaN-B. Details of the explanation are described in Reference 25, which is cited in the Material and Methods (P8, L1).

Overexpression of CaN-B is sufficient to induce K-induced NCC dephosphorylation. We overexpressed CaN-A to try to demonstrate the involvement of CaN.

(2) Figure 2D: why did the authors quantify pNCC-S71 in kidney slices and mice rather than NCC-p53, used in vitro?

Thank you for the comment. We used primary antibodies against both Ser71 and Thr53 to quantify pNCC. Previously, Pacheco-Alvarez et al. [2] reported that both Thr53 and Ser71 sites are critical for NCC regulation. Because both Thr53 and Ser71 are critical phosphorylation sites, many other reports have also evaluated NCC activity by using Ser71 and Thr53 phosphorylation site-specific NCC antibodies [1,3]. They showed that each phosphor-NCC antibody was useful to evaluate NCC activities. Therefore, we used the Ser71 antibody for mouse kidneys and Thr53 antibody for HEK293 cells because these antibodies were useful to detect phopho-NCC specific bands in each case.

- In addition, the figures would be easier to read if the system in which the experiments are conducted were indicated on the figure.　

Thank you for your kind suggestion. We have added a figure of the sliced kidney ex-vivo experiment system in revised S3 Fig.

(3) Figure S11: why did the authors quantify the pNCC/tNCC ratio rather than pNCC and tNCC separately like in the other experiments?

Thank you for the comment. Phosphorylated NCC is an active form of NCC. The pNCC/tNCC ratio is a more suitable parameter to evaluate NCC activity than pNCC/actin. In many previous studies [4-6], the pNCC/tNCC ratio was used to address NCC activity. In the present study, we have shown both the pNCC/tNCC ratio and tNCC/actin ratio. However, the tNCC/actin data was missing from the previous S11 Fig. Therefore, we have added tNCC/actin to the revised S11 Fig. 

 (4) Table 2: ‘Blood K+ level at 120 min following a high-K+ load was significantly higher in SEA0400-treated mice than in vehicle-treated counterparts (Table 2)’.

In their previous study, using tacrolimus to inhibit calcineurin, the authors did not observe a change in plasma potassium in tacrolimus-treated animals even though NCC dephosphorylation was inhibited. Could the authors comment on the difference in plasma K between tacrolimus- and SEA0400-treated animals?

Thank you for the important comment. In our previous study, we analyzed the plasma K level at 90 min after a K load [7]. There was no significant difference in the plasma K level between tacrolimus-treated mice and control mice. Nevertheless, the plasma K level in tacrolimus-treated mice tended to be higher than that in control mice: the plasma K levels in the tacrolimus group and the vehicle group were 6.83 ± 0.50 and 6.23 ± 0.49 [mean ± SEM], respectively, P = 0.2, N = 7). In this study, we tested the plasma K level at 120 min following a high-K load and found that the plasma K level was significantly higher in the SEA0400-treated mice than in the vehicle-treated mice. If we had tested the plasma K level in tacrolimus-tested mice at 120 min, then we would have seen that the plasma K levels in the tacrolimus-treated mice were significantly higher than in the control mice. 

(5) Have the authors studied the role of NCX1 after a longer exposure to high K ?　 

Thank you for the insightful comment. A chronic K load experiment could be more informative. However, the mechanism of NCC regulation with a chronic K load is more complicated than that after an acute K load. As we described in the Introduction (P5, L15), it is necessary to consider the acute and chronic phases separately to understand the effect of high K intake on NCC regulation. In this study, we did not examine the role of NCX1 after a longer exposure to high K. We focused on the acute regulation of NCC phosphorylation. 

Minor comments:

- Abstract : the authors should remove the following sentence, which is not at the correct place : ‘The mice were housed in metabolic cages for urine sample collection’. 

Thank you for the kind suggestion. We have removed the sentence as suggested. 

- Introduction, page 6 : the reference’s number in the following sentence is incorrect: ‘In our previous study, we observed that the CaN inhibitor, tacrolimus, …urinary K+ excretion in the acute phase[22]’. The authors should check all the other references’ number.　

We sincerely apologize for the incorrect citations. We have checked all the reference numbers and corrected them.

- Results: Figures 3 and 5 should be merged.

Thank you for your helpful comment. We have merged the previous Figure 3 and Figure 5 into a revised Figure 3 according to your instruction. We have revised the previous Figure 5 as Figure 3D. 

References:

1. Sorensen MV, Grossmann S, Roesinger M, Gresko N, Todkar AP, Barmettler G, et al. Rapid dephosphorylation of the renal sodium chloride cotransporter in response to oral potassium intake in mice. Kidney Int. 2013;83(5):811-24. 

2. Pacheco-Alvarez D, Cristóbal PS, Meade P, Moreno E, Vazquez N, Muñoz E, et al. The Na+:Cl- cotransporter is activated and phosphorylated at the amino-terminal domain upon intracellular chloride depletion. J Biol Chem. 2006;281(39):28755-63.

3. Kikuchi E, Mori T, Zeniya M, Isobe K, Ishigami-Yuasa M, Fujii S, et al. Discovery of Novel SPAK Inhibitors That Block WNK Kinase Signaling to Cation Chloride Transporters. J Am Soc Nephrol. 2015;26(7):1525-36. 

4. Ferdaus MZ, Barber KW, López-Cayuqueo KI, Terker AS, Argaiz ER, Gassaway BM, et al. SPAK and OSR1 play essential roles in potassium homeostasis through actions on the distal convoluted tubule. J Physiol. 2016;594(17):4945-66.

5. Lazelle RA, McCully BH, Terker AS, Himmerkus N, Blankenstein KI, Mutig K, et al. Renal Deletion of 12 kDa FK506-Binding Protein Attenuates Tacrolimus-Induced Hypertension. J Am Soc Nephrol. 2016;27(5):1456-64.

6. Nomura N, Shoda W, Wang Y, Mandai S, Furusho T, Takahashi D, et al. Role of ClC-K and barttin in low potassium-induced sodium chloride cotransporter activation and hypertension in mouse kidney. Biosci Rep. 2018;38(1).

7. Shoda W, Nomura N, Ando F, Mori Y, Mori T, Sohara E, et al. Calcineurin inhibitors block sodium-chloride cotransporter dephosphorylation in response to high potassium intake. Kidney Int. 2017;91(2):402-11.

---

## [Decision Letter · Decision Letter 1]

15 Jun 2020

Sodium–calcium exchanger 1 is the key molecule for urinary potassium excretion against acute hyperkalemia

PONE-D-19-36071R1

Dear Dr. Nomura,

We’re pleased to inform you that your manuscript has been judged scientifically suitable for publication and will be formally accepted for publication once it meets all outstanding technical requirements.

Kind regards,

Jean-Claude Dussaule

Academic Editor

PLOS ONE

Additional Editor Comments (optional):

Reviewers' comments:

Reviewer's Responses to Questions

**Comments to the Author**

1. If the authors have adequately addressed your comments raised in a previous round of review and you feel that this manuscript is now acceptable for publication, you may indicate that here to bypass the “Comments to the Author” section, enter your conflict of interest statement in the “Confidential to Editor” section, and submit your "Accept" recommendation.

Reviewer #1: All comments have been addressed

Reviewer #2: All comments have been addressed

Reviewer #3: All comments have been addressed

2. Is the manuscript technically sound, and do the data support the conclusions?

Reviewer #1: (No Response)

Reviewer #2: Yes

Reviewer #3: (No Response)

3. Has the statistical analysis been performed appropriately and rigorously? 

Reviewer #1: (No Response)

Reviewer #2: I Don't Know

Reviewer #3: (No Response)

4. Have the authors made all data underlying the findings in their manuscript fully available?

Reviewer #1: (No Response)

Reviewer #2: Yes

Reviewer #3: (No Response)

5. Is the manuscript presented in an intelligible fashion and written in standard English?

Reviewer #1: (No Response)

Reviewer #2: Yes

Reviewer #3: (No Response)

6. Review Comments to the Author

Reviewer #1: (No Response)

Reviewer #2: (No Response)

Reviewer #3: (No Response)

7. PLOS authors have the option to publish the peer review history of their article (what does this mean?). If published, this will include your full peer review and any attached files.

Reviewer #1: No

Reviewer #2: No

Reviewer #3: Yes: Juliette Hadchouel

---

## [Editor Report · Acceptance letter]

18 Jun 2020

PONE-D-19-36071R1 

Sodium–calcium exchanger 1 is the key molecule for urinary potassium excretion against acute hyperkalemia 

Dear Dr. Nomura:

I'm pleased to inform you that your manuscript has been deemed suitable for publication in PLOS ONE. Congratulations! Your manuscript is now with our production department. 

Kind regards, 

on behalf of

Dr. Jean-Claude Dussaule 

Academic Editor

PLOS ONE